# Technical note: "Bit by bit": A practical and general approach for evaluating model computational complexity vs. model performance

Elnaz Azmi[1], Uwe Ehret[2], Steven V. Weijs[3], Benjamin L. Ruddell[4], Rui A. P. Perdigão[5,6,7]

[1]Steinbuch Centre for Computing, Karlsruhe Institute of Technology - KIT, Karlsruhe, Germany
[2]Institute of Water Resources and River Basin Management, Karlsruhe Institute of Technology - KIT, Karlsruhe, Germany
[3]Department of Civil Engineering, University of British Columbia, Canada
[4]School of Informatics, Computing, and Cyber Systems, Northern Arizona University, U.S.A
[5]Meteoceanics Interdisciplinary Centre for Complex System Science, Vienna, Austria
[6]CCIAM, Centre for Ecology, Evolution and Environmental Changes, Universidade de Lisboa, Lisbon, Portugal
[7]Physics of Information and Quantum Technologies Group, Instituto de Telecomunicações, Lisbon, Portugal

*Correspondence to*: Elnaz Azmi (elnaz.azmi@kit.edu)

**Abstract.** One of the main objectives of the scientific enterprise is the development of well-performing yet parsimonious models for all natural phenomena and systems. In the 21st century, scientists usually represent their models, hypotheses, and experimental observations using digital computers. Measuring performance and parsimony of computer models is therefore a
key theoretical and practical challenge for 21st century science. 'Performance' here refers to a model's ability to reduce predictive uncertainty about an object of interest. 'Parsimony' (or complexity) comprises two aspects: descriptive complexity - the size of the model itself which can be measured by the disk space it occupies -, and computational complexity - the model's effort to provide output. Descriptive complexity is related to inference quality and generality, computational complexity is often a practical and economic concern for limited computing resources.

In this context, this paper has two distinct but related goals: The first is to propose a practical method of measuring computational complexity by utility software 'Strace', which counts the total number of memory visits while running a model on a computer. The second goal is to propose the 'bit by bit' method, which combines measuring computational complexity by 'Strace', and measuring model performance by information loss relative to observations, both in bit. For demonstration, we apply the 'bit by bit' method to watershed models representing a wide diversity of modelling strategies (artificial neural
network, auto-regressive, process-based, and other). We demonstrate that computational complexity as measured by 'Strace' is sensitive to all aspects of a model, such as the size of the model itself, the input data it reads, its numerical scheme and time-stepping. We further demonstrate that for each model, the bit counts for computational complexity exceed those for performance by several orders of magnitude, and that the differences among the models for both computational complexity and performance can be explained by their setup, and are in accordance with expectations.

We conclude that measuring computational complexity by 'Strace' is practical, and it is also general in the sense that it can be applied to any model that can be run on a digital computer. We further conclude that the 'bit by bit' approach is general in the sense that it measures two key aspects of a model in the single unit of bit. We suggest that it can be enhanced by additionally measuring a model's descriptive complexity - also in bit.

**Keywords**

Computational complexity, descriptive complexity, model performance, model evaluation, information, entropy

# 1 Introduction

## 1.1  The goals of Science

One of the main objectives of the scientific enterprise is the development of parsimonious yet well-performing models for all
natural phenomena and systems. Such models should produce output in agreement with observations of the related real-world system, i.e. perform well in terms of accuracy and precision and overall 'rightness' (Kirchner, 2006). Another key aspect of evaluating such models is their complexity, i.e. they should be brief, elegant, explainable, understandable, communicable, teachable, and small. Mathematical analytical models - e.g. Newton's Laws - represent an ideal type of model because they combine performance (high accuracy and precision when compared with experimental observations)
with minimal yet adequate complexity (high elegance, brevity, and communicability). Another key aspect of model complexity is how efficiently a model produces its output. This is especially relevant for large models used in operational settings, where computational effort or - closely related - computation times are an issue. In Fig. 1 panel (a), these key aspects of model evaluation are referred to as 'descriptive complexity', 'computational complexity' and 'performance'. A simple example to illustrate their relation: Suppose we want to bake a cake; then the length of the recipe measures its
descriptive complexity, the time or effort it takes to actually prepare the cake by following the recipe instructions measures its computational complexity, and the (dis-) agreement of our cake with the gold standard cake from the pastry shop measures its performance.

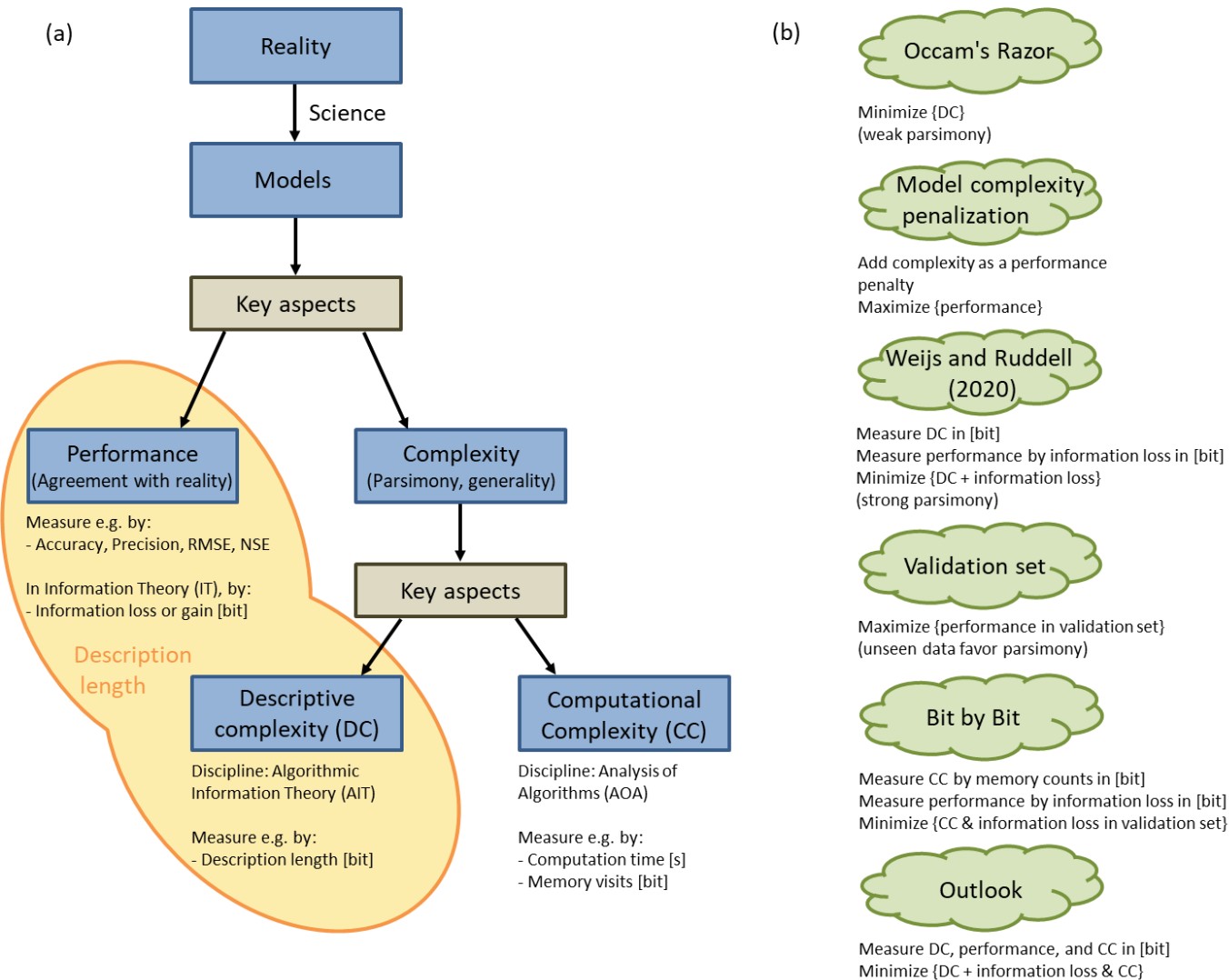

 **Figure 1.** (a): Aspects of model evaluation. (b): Guiding principles for model development.

## 1.2 Guidelines for developing parsimonious models

Many approaches exist to guide model development (Fig. 1 panel (b)), and they differ by the way they handle and the emphasis they put on each of the three previously discussed key aspects (see e.g. Schoups et al., 2008). We will in the

following briefly describe some of these guidelines to provide the background for the 'bit by bit' approach suggested in the paper.

*Occam's Razor*, a bedrock principle of science, argues that the least descriptively complex model is preferable, at a given level of predictive performance that is adequate to the question or application at hand. Occam's Razor is a guideline to

promote models that describe well patterns in the data and to distil laws that allow effective compression of experimental data; also it is a guideline for inference. When applying Occam's Razor, the parsimonious among the well-performing models are identified, but comparisons of models of different complexity for model selection are not possible by this principle alone.

*Model selection by applying complexity penalization measures.* Complexity penalization measures, such as the Akaike Information Criterion (Akaike, 1974) or the Bayesian Information Criterion (Schwarz, 1978), are formalizations of the principle of parsimony which can be applied to make models of varying complexity comparable in terms of performance. Many discipline- and purpose-specific variants for complexity penalization exist, for hydrology, we give a brief overview on definitions of complexity and approaches to build parsimonious models in section 1.4.

In the framework of Algorithmic Information Theory (AIT) (Kolmogorov, 1968; Solomonoff 1964, 1978; Chaitin 1966), descriptive complexity of a model is measured by its size expressed in bit, when stored as instructions for a computer. It is therefore a formalization of Occams razor. Furthermore, the same concept of descriptive complexity can also be directly applied to data. The complexity of data is formalized as its shortest description length, and the best model for the data is that shortest description: the shortest computer program that has the data as an output. It is noteworthy that in all these approaches that employ Occam's razor, an emphasis is placed on descriptive complexity and  performance , but is completely independent of any practical considerations such as limited storage space or computing power, i.e. it ignores computational complexity. So while Occam promotes models that achieve effective compression of experimental data, compression for the sake of meeting constraints in a storage limited world is not the primary goal, but rather the reverse: finding the shortest description is the process of inference, achieved by distilling patterns from data in order to find general predictive laws.

*Weijs and Ruddell (2020),* call Occam's parsimony a 'weak parsimony' because it identifies a set of parsimonious models rather than a single most-parsimonious model. They further argue that a single, 'strongly parsimonious' model could be identified by considering, in addition to model descriptive complexity, also model performance, and to express them in the language of AIT as two additive terms which together are the description length of the data in bits (yellow shaded area in Fig. 1 panel (a)). Performance thus becomes part of parsimony by collapsing them in the single dimension of description length. A strongly parsimonious model in the terms of Weijs and Ruddell (2020) perfectly (or losslessly) reproduces experimental observations in the smallest number of bits, after adding together the compressed size of the model and the compressed corrections needed to adjust the model's predictions to equal the observations. Such a model balances minimum model size and minimum information loss, and maximum generalizability outside the observed datasets used to construct and test the model. The latter claim is based on insights from AIT, where shorter descriptions have been shown to be more likely to be generalizable. This is expressed through the concept of algorithmic probability, assigning higher prior probability to simple models, and convergence of induction systems based on this formalization was shown in Solomonoff (1978). Another way to see this is that by finding the minimum description length, all structure in the data is exploited. This was used by Kolmogorov (1968) to define randomness as absence of structure and therefore as incompressibility. Detailed explanations on this topic are given in Weijs and Ruddell (2020) and references therein. The approach proposed by Weijs

and Ruddell (2020), drawing on the minimum description length principle (Rissanen, 1978; Grünwald, 2007) not only has the advantage of favouring models with a good trade-off between descriptive complexity and performance: Applying a single measure, expressed in bits, to quantify both of these aspects also offers the advantage of rigor and generality over more contextually defined performance measures, such as Root Mean Square Error, Nash-Sutcliffe Efficiency (Nash and Sutcliffe, 1970), Kling-Gupta Efficiency (Gupta et al., 2009), to name just a few (more in Bennett et al., 2013). This more generalized strategy helps guiding model preference, especially in automated environments for learning models from data, starting with the widest class of all computable models, and making very few prior assumptions on structure. At the same time, the lack of prior assumptions is also a weakness of this framework in contexts where considerable prior information is available. In hydrology, this is typically the case, therefore, practical application of this framework is still an open challenge.

*Validation set* approaches are a standard procedure in hydrological model development. Among a set of competing models, the model is preferred that performs best on data unseen during model parameter estimation. The fact that model performance is evaluated on a validation set promotes models that are general, i.e. models that have captured the essential workings of the natural system they represent, and demotes models overfitted to the calibration data, which are likely to be models with unnecessarily high descriptive complexity. It is therefore an implicit form of model complexity control.

In summary, both Occam's razor and the AIT-based extension argued for by Weijs and Ruddell (2020) are designed with a focus on inference, i.e. on distilling small and universal laws from experimental data, while the focus of validation set approaches is mainly on performance. In neither of them the model's effort of actually making its predictions is directly considered. This effort, however, can be an important quality of a model in settings where computing resources are limited. In earth science modelling, this is the rule rather than the exception for the following reasons: i) scales of earth systems cannot be separated easily and in some cases not at all, so even for local questions it may be necessary to simulate large systems at a level of great spatio-temporal detail; ii) calibration of model parameters from data needs many repeated model runs for parameter identification; iii) models used in optimal decision making require repeated use to identify the optimal alternative. The efficiency at which models generate their output is subject of the discipline of Analysis of Algorithms (AOA). In AOA, it is referred to as computational complexity, and can be measured in terms of two finite resources that are needed for computation: time and/or space. Time-complexity relates to the time a computer needs to arrive at the result. Time complexity can be measured in terms of clock cycles or number of floating point operations, and often it is the scaling with the input size that is of interest. Space-complexity relates to the memory used, i.e. the maximum number of binary transistor states needed during the execution of the program. As for descriptive complexity, the reads of this memory can be interpreted as answers to Yes/No questions, and can be measured in bit.

### 1.3 Scope and goals of this paper

In the context of the guidelines for model development discussed in the previous section, this paper has two distinct but related goals: The *first goal* is to propose a practical method of measuring computational complexity by 'Strace', a troubleshooting and monitoring utility for computer programs. 'Strace' counts the total number of memory visits while

running a model on a computer. The counting is sensitive to all aspects of the model, such as the size of the model itself, the size of the input data it reads, the model's numerical scheme, time-stepping, runtime environment, etc. The *second goal* is to demonstrate how measuring computational complexity by 'Strace' can be combined with either a validation set approach or the approach suggested by Weijs and Rudell (2020) to jointly evaluate all key aspects of a model - descriptive complexity, computational complexity, and performance. We use a validation set approach here, as hydrologists are familiar with it, but adopt from Weijs and Ruddell (2020) to express model performance by information loss in bit. The 'bit by bit' approach as presented here therefore consists of explicitly evaluating a model in terms of computational complexity and performance, both in bit. Descriptive complexity is implicitly considered by the validation set approach. Measuring computational complexity by 'Strace' is general in the sense that it can be applied to any model that can be run on a digital computer; the 'bit by bit' approach is general in the sense that it measures two key aspects of a model in the single unit of bit.

For demonstration, we run hydrological models of various types (artificial neural network, auto-regressive, simple and more advanced process-based, and both approximate and exact restatements of experimental observations) that all aim to perform the same task of predicting discharge at the outlet of a watershed. Akin to Weijs and Ruddell (2020), we examine possible trade-offs between computational complexity vs. information loss. It is important to note that the purpose of the model comparison here is not primarily to identify the best among the different modelling approaches, rather it serves as a demonstration how 'Strace' is sensitive to all facets of a model, and how differences among the models can be explained by their setup and are in accordance with expectations. In short, the aim is to provide a proof-of-concept.

The remainder of the manuscript is structured as follows: In section two, we describe the real-world system we seek to represent (a small alpine watershed in western Austria), the range of models we use for demonstration of the 'bit by bit' concept, and the implementation environment and criteria for measuring model performance and computational complexity. In section three, we present and compare the models in terms of these criteria and illuminate differences between descriptive and computational complexity. In section four, we draw conclusions, discuss the limitations of the approach, and provide directions for future work.

## 1.4 Uses of 'complexity' in the hydrological sciences

A brief note on the uses of the term 'complexity', in this paper and in the hydrological sciences in general: In this paper, we use it in very specific ways to refer to different characteristics of a model. We have adopted the term 'descriptive complexity' from Algorithmic Information Theory to express the parsimony of a model by its size in bit when stored on a computer, and the term 'computational complexity' from Analysis of Algorithms to express the efficiency at which a model generates its output by the number of memory visits during program execution. In the hydrological sciences in general, 'complexity' is most often used in the wide sense of its dictionary definition to refer to 'systems consisting of many, different but related parts that are hard to separate, analyse, explain or understand' (see also Gell-Mann, 1995 on various interpretations of complexity). Hydrological systems have been described and analysed in terms of their complexity by Jenerette et al. (2012), Jovanovic et al. (2017), Ossola et al. (2015), Bras (2015) and others; similarly, hydrological time series complexity was

investigated by e.g. Engelhardt et al. (2009). Complexity measures have been used for classification of hydrological systems by Pande and Moayeri (2018), who used the Vapnik-Chervonenkis dimension from statistical learning theory (Cherkassky and Mulier, 2007). This is yet another view on model complexity as its flexibility to classify arbitrary data. Other complexity based classification was done by Sivakumar and Singh (2012) and Sivakumar et al. (2007). In this context, many different complexity measures have been proposed based on e.g. information entropy (Zhou et al., 2012; Castillo et al., 2015), wavelets (Sang et al., 2011), correlation dimension of system output (Sivakumar and Singh, 2012), and dynamic source analysis (Perdigão 2018; Perdigão et al. 2019). In hydrological modelling, 'model complexity' most often refers to the number of processes, variables, or parameters a model comprises, and many authors have investigated the relation of model complexity and predictive performance (Gan et al., 1997; Schoups et al., 2008; Arkestein and Pande, 2013; Forster et al., 2014; Finger et al., 2015; Orth et al., 2015) and proposed ways to build or select models of minimally adequate complexity, i.e. parsimonious models (Atkinson et al., 2002; Sivapalan, 2003; McDonnell et al., 2007; Schöninger et al., 2015; Höge et al., 2018).

Our research contributes to the large existing body of complexity studies in hydrology, and we believe that by expressing all key aspects of computer-based models – performance, descriptive complexity and computational complexity – in the single general unit of bit can help facilitating comprehensive model evaluation and optimization.

## 2 Methods

### 2.1 The real-world system: a watershed in Austria

The real-world system we seek to represent with our models is the Dornbirnerach catchment in western Austria. Upstream of river gauge Hoher Steg (Q_Host), the target of our model predictions, the catchment covers 113 km². The catchment's rainfall-runoff dynamics reflect its alpine setting: Winter snow accumulation, spring snowmelt, high and intensive summer rainfall and, due to the steep terrain, rapid rainfall-runoff response. The meteorological dynamics of the system are represented by precipitation observations at a single rain gauge, Ebnit (P_Ebnit), located in the catchment centre. Both time series are available in hourly resolution for ten years (1996/01/01 – 2005/12/31). No other dynamical or structural data were used for model set up. While this would be overly data-scarce if we wanted to build the best possible hydrological model, we deemed it adequate for the aim of this study, i.e. demonstration of the bit-by-bit approach.

### 2.2 Models

We selected altogether eight modelling approaches with the aim of covering a wide range of model characteristics such as type (ignorant, perfect, conceptual-hydrological and data-driven), structure (single and double linear reservoir), numerical scheme (explicit and iterative) and precision (double and integer). The models are listed and described in Table 1, additional information is given in Fig. 2. We trained/calibrated each model on a five-year calibration period (1996/01/01 – 2000/12/31) and validated them in the five-year validation period (2001/01/01 – 2005/12/31).

**Table 1.** Models used in the study and their characteristics.

| ID | Description | Time stepping $dt$ | Variable precision | Numerical scheme | Training data | Data for running the model |
|---|---|---|---|---|---|---|
| Model-00 | An (almost) ignorant model, which predicts for each time step the observed time series mean (4.86 m³/s) | 1 h | double | -- | Q_Host | -- |
| Model-01 | A perfect model representing full prior knowledge contained in the experimental observations. For each time step, the observed value of Q_Host is read as input and provided as output. | 1 h | double | -- | Q_Host | Q_Host($t$) |
| Model-02 | A simple conceptual hydrological model, representing the catchments' rainfall-runoff behaviour by a single linear reservoir (Fig. 2, panel (a) ) with a single parameter - K -, and a single state variable - S. K was found by calibration (K = 64 h). Time stepping is d$t$ = 1 h, all variables are double precision, and the numerical scheme is explicit. | 1 h | double | explicit | Q_Host P_Ebnit | P_Ebnit ($t$) |
| Model-02a | Same as Model-02, but K is an uncalibrated initial value (K = 120 h) | 1 h | double | explicit | Q_Host P_Ebnit | P_Ebnit ($t$) |
| Model-02b | Same as Model-02, but time stepping is d$t$ = 1 min | 1 min | double | explicit | Q_Host P_Ebnit | P_Ebnit($t$) |
| Model-02c | Same as Model-02, but all variables are integer precision only | 1 h | integer | explicit | Q_Host P_Ebnit | int(P_Ebnit($t$)) |
| Model-02d | Same as Model-02, but the numerical scheme is iterative. | 1 h | double | iterative | Q_Host P_Ebnit | P_Ebnit(t) |
| Model-03 | A more advanced conceptual model (Fig. 2, panel (b) ). Precipitation input is split by an intensity threshold - T - (3.5 mm/h), and enters two linear reservoirs - K1 - (10 h) and - K2 - (80 h). All parameters were found by calibration. Time stepping, variable precision, and numerical scheme are the same as in Model-02. | 1 h | double | explicit | Q_Host P_Ebnit | P_Ebnit($t$) |
| Model-04 | A long short-term memory artificial recurrent neural network (LSTM) with a single hidden layer of 5 neurons and rolling window of size 20 along the time axis, using P_Ebnit($t$) as input to predict Q_Host($t$). The model is written in Python with the 'Keras' library. In the learning process, it uses the 'adam' optimizer with the loss function 'mean squared error'. | 1 h | double | -- | Q_Host P_Ebnit | P_Ebnit(t) |
| Model-05 | A simple third-order autoregressive model, which predicts Q_Host($t$) by a linear combination of previous observations in the form $Q(t) = c_0 + c_1 Q(t-1) + c_2 Q(t-2) + c_3 Q(t-3)$ All coefficients were found by calibration ($c_0 =$ | 1 h | double | -- | Q_Host | Q_Host($t$-3) Q_Host($t$-2) Q_Host($t$-1) |

| | 0.0536, $c_1 = 1.9916$, $c_2 = -1.3130$, $c_3 = 0.3104$). Testing models of various order we found that adding observations beyond lag-3 improved predictive power only marginally | | | | | |
|---|---|---|---|---|---|---|
| Model-06 | An artificial neuronal network (ANN) with a single hidden layer of 5 neurons, using Q_Host($t$-1, $t$-2, $t$-3) to predict Q_Host($t$). The model is written in Python with the 'Keras' library. In the learning process, it uses the 'adam' optimizer with the loss function 'mean squared error'. | 1 h | double | -- | Q_Host | Q_Host($t$-3) Q_Host($t$-2) Q_Host($t$-1) |

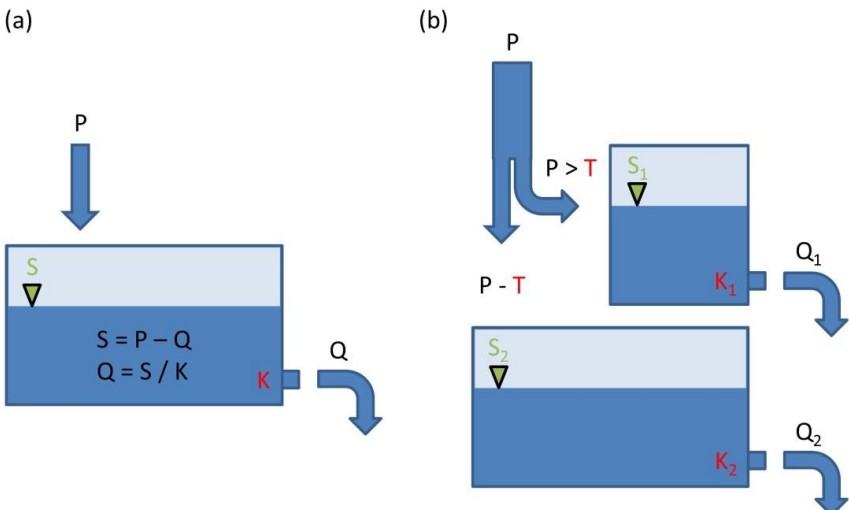

**Figure 2.** (a): Model-02, a single linear reservoir with state variable S and retention constant K. The reservoir is replenished by precipitation P and drained by discharge Q. (b): Model-03, with two linear reservoirs. Precipitation input is split by intensity threshold T.

### 2.3 Implementation environment

All models were implemented as Python scripts running on Python 3.6 with the installed packages Numpy, Pandas, Scipy, Keras and H5py. The experiments were done on a computer running Red Hat Enterprise Linux Server release 7.4 on a 16-core Intel(R) Xeon(R) CPU E5-2640 v2 @ 2.00 GHz processor.

### 2.4 Measures of model performance and computational complexity

All models were evaluated in terms of the two criteria described in the introduction: performance, i.e. the model's ability to reduce predictive uncertainty about the target, and computational complexity, i.e. the effort required to make the model generate a prediction about the target. Similar to Weijs and Ruddell (2020), we express both quantities in bits, to be able to investigate whether direct comparison or combining both counts in a single measure helps interpretation.

### 2.4.1 Model performance

As in Weijs and Rudell (2020), we express model performance in terms of information losses. In information theory, information is defined as the negative logarithm of the probability $p$ of an event. Information entropy $H(X)$ is defined as the expected or average value of information (Eq. 1) of a specific value of a data set $X = \{x_1, x_2, ..., x_n\}$.


$$H(X) = \mathbf{E}[I(x)] = -\sum_{x \in X} p(x) \, log_2 \, p(x) \tag{1}$$

Entropy is a measure of our uncertainty about the outcome of a random draw from a distribution *before* it is revealed to us, when all we know a priori is the data distribution. If we know the outcome beforehand (e.g. because we cheated), then the a priori known data distribution reduces to a Dirac function with $p = 1$ for the outcome and $p = 0$ everywhere else. The

entropy of such a distribution - and with it our uncertainty - is zero. In model performance evaluation, we can use this 'perfect prediction' case as a benchmark to compare other states of prior knowledge against in terms of added uncertainty (or information lost). In the case described above, where all we know a priori is the data distribution, the information loss compared to the benchmark case equals the entropy of the distribution. In other cases, we may have useful side information – e.g. predictions of a model, which reduces information loss. In such a case, information loss can be quantified by conditional

entropy (Eq. 2), where $X$ represents the target and $Y$ the model predictions (= predictor), $y$ is a particular prediction, and $H(X/Y)$ is conditional entropy in bit.

$$H(X|Y) = \sum_{y \in Y} p(y) \, H(X|Y = y) = -\sum_{y \in Y} p(y) \sum_{x \in X} p(x|y) \, log_2 \, p(x|y) \tag{2}$$

Note that for models providing single-valued (deterministic) predictions, in order to construct a predictive distribution for

which we can calculate an entropy, we have to assume that our state of knowledge for each prediction is given by the subset of observations jointly occurring with the particular prediction (the conditional distribution of $X$ for a particular $y$). If models would give probabilistic predictions, we could directly employ a relative entropy measure such as Kullback-Leibler divergence (Kullback and Leibler, 1951; Cover and Thomas, 2006), which would lead to fairer assessments of information loss (Weijs et al., 2010). However, models that directly output probabilistic predictions are not yet a standard in hydrology.

Alternatively to measuring information losses of model predictions compared to an upper benchmark - the observations – as described above, it is also possible to measure information gains compared to a lower benchmark – the entropy of a uniform distribution - which expresses minimum prior knowledge. Weijs and Ruddell (2020), which we refer to throughout the text, used information losses because they directly translate to a description length. For reasons of comparability we applied the same concept here.

To avoid fitting of theoretical functions to the empirical data distributions, we calculated conditional entropy of discrete (binned) distributions, i.e. normalized empirical histograms. Choice of the binning scheme has important implications on the values of the information measures derived from the binned distributions: While the lower bound for entropy, $H = 0$ for a Dirac distribution, is independent of the number of bins $n$, the upper bound, $H = \log_2(n)$ for the maximum-entropy uniform distribution is a function of $n$. Choice of $n$ is typically guided by the objective to balance resolution and sufficiently

populated bins, and different strategies have been proposed e.g by Knuth (2013), Gong et al. (2014) and Pechlivanidis et al. (2016). In this context, several estimators for discrete distributions based on limited samples have been proposed that both converge asymptotically towards the true distribution and at the same time provide uncertainty bounds as a function of sample size and binning choice. In Darscheid et al. (2018), both a Bayesian approach and a Maximum-Likelihood approach are presented. We applied uniform binning as it introduces minimal prior information (Knuth, 2013) and as it is simple and

computationally efficient (Ruddell and Kumar, 2009). We uniformly split the value range of 0 - 150 m³/s, which covers all observed and simulated values of Q_Host (0.05 – 137 m³/s) into 150 bins of 1 m³/s width each. Compared to the typical error associated with discharge measurements in small, alpine rivers, which may be as high as 10%, we deemed this an adequate resolution which neither averages away the data-intrinsic variability nor fine-grains to resolutions potentially dominated by random errors.

When calculated in the described manner, a lower bound and two upper benchmarks for the values of conditional entropy can be stated: If the model perfectly predicts the true target value, it will be zero. Non-zero values of conditional entropy quantify exactly the information lost by using an imperfect prediction. If predictor and target are independent, the conditional entropy will be equal to the unconditional entropy of the target, which in our case is H(Q_Host) = 3.46 bit. If in the worst case there would be no paired data of target and predictors to learn from via model calibration, and the physically

feasible range of the target data would be the only thing known a priori, the most honest guess about the target value would be a uniform (= maximum entropy) distribution. For the 151 bins we used, the entropy of a uniform distribution is $H_{uniform} = \log_2(151) = 7.23$ bit.

### 2.4.2 Model computational complexity

    We quantify computational complexity by the total number of memory read and write visits (in bit) on a computer while

running the model. In the context of Information Theory, these bit counts and the bits measuring model performance by conditional entropy in the previous section can both be interpreted in the same manner as a number of binary Yes/No questions that were either already asked and answered during the model run (in the former case) or still need to be asked (in the latter case) in order to fully reproduce the data.

    Counting memory visits while running a computer program can be conveniently done by 'Strace', a troubleshooting and

monitoring utility for Linux (see http://man7.org/linux/man-pages/man1/strace.1.html). It is a powerful tool to diagnose, debug and trace interactions between processes and the Linux kernel (Levin and Syromyatnikov, 2018). 'Strace' is executable along with running code in any programming language like python, C ++ or R. We instructed 'Strace' to monitor our test

models written in Python by counting the total number of bytes read during the model execution from a file stored in the file system into a buffer, and the total number of bytes written from a buffer into a file stored in the file system. A buffer is a temporary data storing memory (usually located in the RAM) that prevents I/O bottleneck and speeds up the memory access. These counts reflect the entire effort of the model to produce the desired output: Reading input files, writing output files, reading the program itself and all system functions called during its execution, efforts of numerical iteration within the program as well as efforts to read and write state variables during runtime. Hence, 'Strace' will penalize models which require large amounts of forcing data, run on high-resolution time stepping or spatial resolution, or apply unnecessarily high-iterative numerical schemes. In short, 'Strace' evaluates all memory-related components of a model in the widest sense.

To evaluate the reproducibility of the countings, we repeated each model run 100 times, clearing the memory cache between individual runs. As the countings were in fact very close, we simply took the average of all runs as a single value representing model computational complexity. The main steps of applying 'Strace' in our work were as follows:

1. We traced the read() and write() system calls of the models while executing their code in Python and wrote them into a target log file running the following command in Linux commandline: 'strace -o target.log -e trace=read/write python model.py', where 'strace' is the executable tool, '-o target.log' is the option to set our log file path, '-e trace=read' traces the read() system call that returns the number of bytes read from the required files during the model execution into the system buffer, 'python' is the path to executable python program and 'model.py' is the path to our model code. Additionally, we used '-e trace=write' to trace the write() system call that returns the number of bytes written from the system buffer into the output file.

2. After generating the target log file, we calculated the sum of all read operations from the target log file running the following command: 'cat target.log | awk 'BEGIN {FS="="}{sum += $2} END {print sum >> "read_sum.txt"}', where 'cat' reads the target.log file, 'awk' scans the file and sums up the returned value of each read() and writes in the 'sum' variable, 'print sum' writes the sum value into a file called 'read_sum.txt'. Similarly, we summarized all write operations in a file. The sum of these read and write values is the total number of bytes which presents the evaluation of our model.

**3 Results and discussion**

As stated previously, it is *not* the primary purpose of the model comparison presented here to identify the best among a set of competing models for a particular purpose. Rather, it is intended as a demonstration and proof-of-concept of how 'Strace' is sensitive to all facets of a model, how 'Strace' and the 'bit by bit' concept are applicable to a wide range of modelling approaches, and how they might be used to guide model optimization and model selection. We do so for six use cases - described in detail in section 3.2 - representing different steps along the iterative process of model building and evaluation as described in Gupta et al. (2008).

### 3.1 Simulation vs. experimental observation

Before discussing the model results for the six use cases in terms of performance and computational complexity, we first provide a short and exemplary visualization of the model predictions to illustrate their general behaviour. In Fig. 3, observed precipitation at Ebnit and observed and simulated discharge time series of all models at gauge Hoher Steg are shown for a rainfall-runoff event in June 2002, which lies within the validation period. The observed hydrograph (bold blue) shows a flood peak of 71 m³/s due to a 14-hour rainfall event. The ignorant Model-00 (black) is incapable of reproducing these dynamics and remains at its constant mean value prediction. Model-01 (light green) as expected perfectly matches the observations, and likewise the AR-3 Model-05 (dark green) and the ANN Model-06 (red) show almost perfect agreement. The single-bucket Model-02 (purple) overall reproduces the observed rise and decline of discharge, but fails in the details: The rise is too slow and too small, and so is the decline. Apparently, a single linear reservoir cannot adequately represent the catchment's hydrological behaviour, irrespective of the time stepping and the numerical scheme: Discharge simulations by the high-resolution Model-02b (pink) and the iterative Model-02d (pink) are almost identical to that of Model-02. Model calibration and data precision however do play a role: The uncalibrated Model-02a (pink dashed) shows clearly worse performance than its calibrated counterpart Model-02, and so does Model-02c (pink dashed-dotted), identical to Model-02 except for a switch from double to integer precision for all variables. For Model-02c, the hydrograph is only coarsely reproduced by a two-step series. From all bucket models, the two-bucket Model-03 (brown) performs best, correctly reproducing the overall course of the event. The LSTM Model-04 (light blue) also provides a good representation of the event rise, recession and peak discharge magnitude, but shows a delayed response with a lag of about three hours.

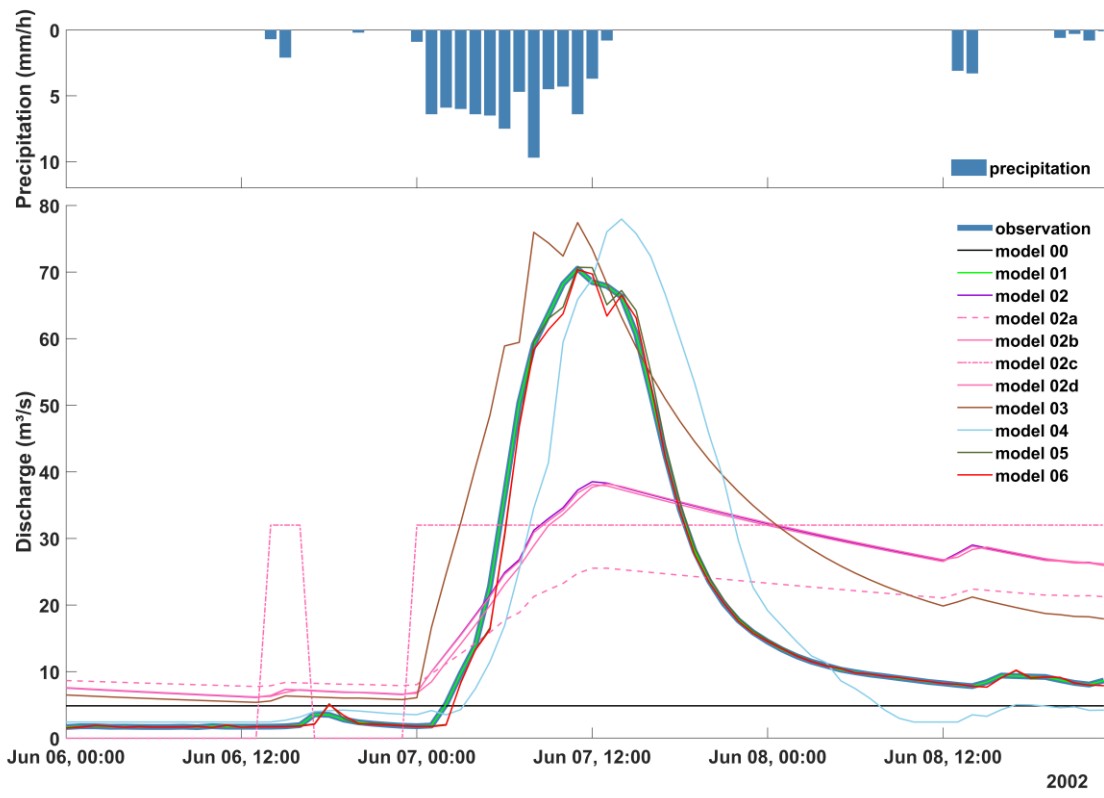

**Figure 3.** Top: Observed precipitation at Ebnit. Bottom: Observed discharge at gauge Hoher Steg and simulations thereof by Model-00 to Model-06 for a rainfall-runoff event in June 2002.

325

## 3.2 Performance vs. computational complexity

Here we discuss the model results in terms of model performance and model computational complexity for six use cases. Model performance is expressed as the remaining uncertainty, at each time step, about the observed data D given the related model simulation M by conditional entropy H(D|M) as described in section 2.4.1. Model computational complexity is

330 expressed as the total number of memory read and write visits during model execution as counted by 'Strace'. For easier interpretation, we show average computational complexity per time step by dividing the total number of visits by the length of the validation period (43802 time steps). Fig. 4 shows computational complexity and performance of all models. The theoretical optimum of zero information loss despite zero modelling effort lies in the lower left corner.

*Use case 1* compares the simple bucket Model-02 to benchmark Model-00 and Model-01 to provide a perspective on the

335 range of possible performance results. In terms of computational complexity, the models differ only slightly (Model-00 1776

bits, Model-02 1797 bits, Model-01 1808 bits), the main difference lies in model performance: As to be expected, Model-01, which simply reproduces the observations, shows perfect model performance (zero information loss). The mean Model-00, also as to be expected shows the worst performance of all models. Taken together, these two models provide a background against which other models can be placed in terms of performance. The single-bucket Model-02 - our standard model - for example shows better performance than the mean model, but is still far from being perfect.

*Use case 2* compares two versions of the simple bucket model: Model-02 is calibrated (K = 64 h, see Table 1), Model-02a is uncalibrated, with the value of K set to a reasonable default value of 120 h. This use case corresponds to a situation during model calibration, where the conceptual model is fixed, and optimal parameters are determined by parameter variation. The two models are equal in terms of computational complexity, but their performance difference (2.8 bits for the first, 2.88 bits for the latter) reveals the benefit of calibration. This shows that model performance expressed by information loss can be used as an objective function during model calibration in a validation set approach (see section 1.2).

*Use case 3* again applies the simple bucket Model-02, but this time it is compared to variations thereof in terms of time stepping (Model-02b), variable precision (Model-02c), and numerical scheme (Model-02d). This use case corresponds to a situation where an adequate numerical model for a given conceptual and symbolic model is sought. Increasing temporal resolution (Model-02b) only increases computational complexity (from 1797 bits to 4217 bits) but has no effect on performance. Obviously, for the given system and data, hourly time stepping is adequate. Variable precision is important in terms of performance: Model-02c, using integer precision variables, performs clearly worse than Model-02, it even performs worse than the uncalibrated Model-02a. The related computational savings - 1755 bits for Model-02c instead of 1797 bits for Model-02 - are small. Despite our expectations, implementing an iterative numerical scheme (Model-02d) has almost no effect on both performance and computational complexity. Investigating the iterative model during runtime revealed that for the hourly time stepping, results were usually stable at first try, such that on average only 1.8 iterations per time step were required, increasing computational complexity only from 1797 bits (Model-02) to 1798 bits (Model-02d). The reason lies in the pronounced autocorrelation of the hydrological system response, such that in just a few cases - mainly at the onset of floods - iterations were actually needed to satisfy the chosen iteration precision limit of 0.001. For the models used here, the effect of the numerical solver on computational complexity is small, however for other systems and models, this can be more important. Overall, this use case demonstrates how different numerical implementations of the same model can be evaluated with the bit by bit approach, which can be helpful when both performance and computational effort of a model are important, e.g. for global-scale, high-resolution weather models.

*Use case 4* compares the simple bucket Model-02 with the more advanced two-bucket Model-03. This corresponds to a situation where a modeller compares competing process hypotheses formulated within the same modelling approach (here: conceptual hydrological models). The two-bucket model performs better (2.76 bits instead of 2.80 bits), at the cost of increasing computational complexity from 1797 bits to 1798 bits. Given this small computational extra cost, a user will likely prefer the conceptually advanced Model-03 here.

*Use case 5* represents a situation of comparing competing modelling approaches, here the conceptual bucket Model-02 and the LSTM Model-04. Both models take the same input – precipitation – to assure comparability. Interestingly, the bucket model here not only performs better than the LSTM (2.80 bits vs. 3.03 bits), it is also much more efficient: The LSTM's computational complexity is almost three times higher than that of the bucket model (1797 bits vs. 6083 bits). Here, the obvious choice for a modeller is the bucket Model-02.

*Use case 6* also compares competing modelling approaches – the autoregressive Model-05 and the neuronal network Model-06 – but this time the models use previously observed discharge as input. Both models make good use of the high information content in this input, such that their performance (0.67 bit for Model-05, 0.68 bit for Model-06) is much better than for all other models except the perfect Model-01. However, they differ in computational complexity: The autoregressive Model-05 only requires 1817 computational bits, while the ANN Model-06 requires 5971 bits. As the autoregressive Model-05 performs better and does so more efficiently, a modeller will likely prefer this over the neuronal network Model-06.

In the lower left corner of Fig. 4, a black square indicates a loose upper bound of the descriptive complexity of a single recording of our target discharge series Q_Host. The value (18.8 bit) was calculated by simply dividing the size of the Q_Host validation dataset by the number of time steps. This represents the raw size of a single data point in the series, without any compression, and if we want we can compare it to the computational effort of *generating* a single data point by any of the models. Clearly, the descriptive complexity is much smaller than the computational complexity.

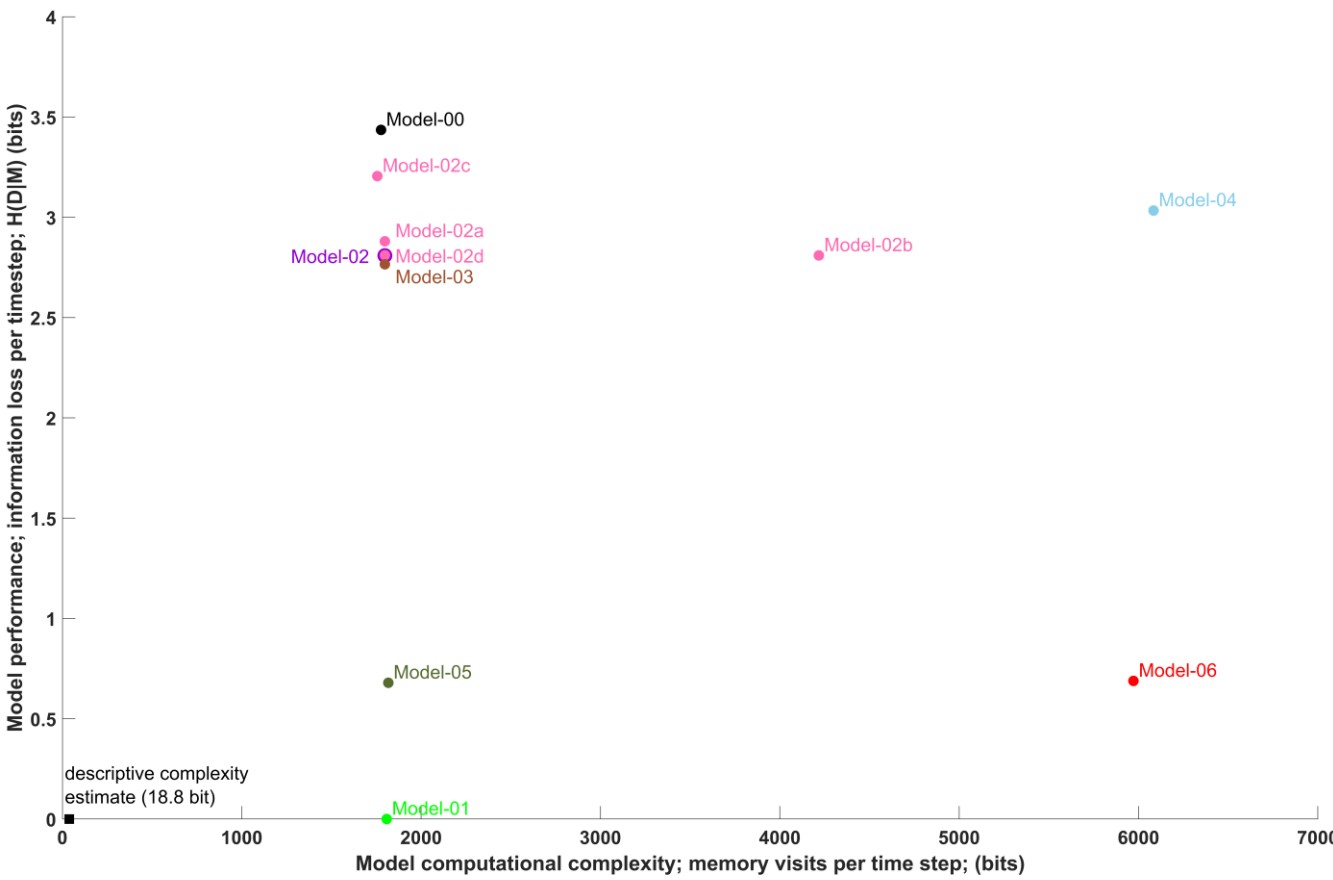

**Figure 4.** Model performance expressed by its inverse, information loss per time step, measured by conditional entropy in bits vs. model computational complexity measured by the average number of memory visits per time step in bits for Model-00 to Model-08.

## 4 Summary and conclusions

We started this paper by stating that one of the main objectives of the scientific enterprise is the development of well-performing yet parsimonious models for natural phenomena and systems, that models nowadays are mainly computer models, and that three key aspects for evaluating such models are descriptive complexity, computational complexity, and performance. We continued by describing several paradigms to guide model development: Occam's razor puts an emphasis on descriptive complexity, and is often combined with performance considerations, but it ignores computational complexity; Weijs and Ruddell (2020) express both model performance and descriptive complexity in bit, and by adding the two obtain a single measure for what they call 'strong parsimony'; validation set approaches focus on performance, and promote general and parsimonious models only indirectly by evaluating models on data not seen during calibration. Neither of these approaches directly incorporates computational complexity. We suggested to close this gap by 'Strace', a troubleshooting and

monitoring utility, which measures computational complexity by the total number of memory visits while running a model on a computer. We further proposed the 'bit by bit' method, which combines measuring computational complexity by 'Strace', and measuring model performance by information loss relative to observations, all in bit, akin to Weijs and Ruddell (2020).

For a proof-of-concept, we applied the 'bit by bit' method in combination with a validation set approach - to also consider
descriptive complexity, if only indirectly - at the example of a range of watershed models (artificial neural network, autoregressive, simple and advanced process-based with various numerical schemes). From the tested models, a third-order autoregressive model provided the best trade-off between computational complexity and performance, while the LSTM and a conceptual model operating in high temporal resolution showed very high computational complexity. For all models, computational complexity (in bit) exceeded the missing information (in bit) expressing model performance by about three
orders of magnitude. We also compared a simple upper bound of descriptive complexity of the target data set to model computational complexity: The latter exceeded the former by about two orders of magnitude. Apart from these specific results, the main take-home messages from this proof-of-concept application are that i) measuring computational complexity by 'Strace' is general in the sense that it can be applied to any model that can be run on a digital computer; ii) 'Strace' is sensitive to all aspects of a model, such as the size of the model itself, the input data it reads, its numerical scheme and time-
stepping; iii) the 'bit by bit' approach is general in the sense that it measures two key aspects of a model in the single unit of bit, such that they can be used together to guide model analysis and optimization in a Pareto trade-off manner in the general setting of incremental learning. It can be useful especially in operational settings where the speed of information processing is a bottleneck. Unlike approaches to estimate computational complexity via model execution time, the bit counting by 'Strace' is unaffected by other ongoing processes on the computer competing for CPU time. This increases reproducibility
and unambiguousness of the results. The 'bit by bit' approach can help promoting better model code in two ways: Computational complexity is sensitive to poor (inefficient) coding, performance is sensitive to wrong (erroneous) coding. This is relevant as computer models in the earth sciences have grown increasingly complex in recent years, and efficient, modular, and error-free code is a precondition for further progress (Hutton et al., 2016).

During the development of this paper we encountered several interesting – and still open – questions: The first was about
where to set the system boundaries: For example, should forcing data – which often are key drivers of model performance - be considered part of the model and hence be included into the counting, or not? If we consider a model that performs well even with limited input data to be more parsimonious than another, which heavily relies on information contained in the input, we should do so. But we could also argue that the input is not part of the model, and should therefore be excluded from the counting. This question also applies to the extent to which the computational setting on the computer should be
included into the counting, and is open for debate. We also still struggle to provide a rigorous description of the nature and strength of the relation between descriptive and computational complexity. Clearly they describe two distinctly different characteristics of a model, but they are also related, as 'Strace' counts both the size of a program and the computational effort of running it. Like performance measured by information loss, the descriptive complexity of a model is typically orders of

magnitude smaller than its computational complexity, which renders their simple additive combination to a single, overall

measure of model quality impractical. Nevertheless, we suggest that combining the approach by Weijs and Ruddell (2020) with measuring computational complexity by 'Strace' will be worth exploring in the future. It potentially offers a comprehensive and multi-faceted way of model evaluation applicable across the earth sciences, where all key aspects of a model are expressed in a single unit, bit.

*Code and data availability.* The code and data used to conduct all analyses in this paper and the result files are publicly available at https://github.com/KIT-HYD/model-evaluation (last access: 2020/03/03).

*Author contributions.* EA wrote all Python scripts and code related to Strace and conducted all model runs. UE designed the study and wrote all Matlab scripts to calculate conditional entropies. EA, UE, SW, BR and RP wrote the manuscript
together.

*Acknowledgements.* We want to thank Jörg Meyer from KIT-SCC for helpful discussions about the best way to bit-count models, Markus Götz from KIT-SCC for discussions about the LSTM model, and Clemens Mathis from Wasserwirtschaft Vorarlberg, Austria, for providing the case study data. We acknowledge support by Deutsche Forschungsgemeinschaft DFG
and Open Access Publishing Fund of Karlsruhe Institute of Technology (KIT). This research contributes to the 'Catchments As Organized Systems" (CAOS) research group funded by the Deutsche Forschungsgemeinschaft DFG. RP also acknowledges FCT under references UIDB/00329/2020 and UID/EEA/50008/2019. BR acknowledges Northern Arizona University for providing start up grants used in part for this work.

*Competing interests.* The authors declare that they have no conflict of interest.

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
