# Peer review of "Technical note: "Bit by bit": A practical and general approach for evaluating model computational complexity vs. model performance"

_Hydrology and Earth System Sciences, 2020_

## Short Comment (SC1) · 10 Apr 2020

**Suggestions to expand the list of candidate models**

I enjoy reading this Discussion paper from a hydrologic model performance perspective, and suggest the authors consider expanding the list of candidate models (Table 1) and their training methodology as follows:

1. A second–order autoregressive process as a baseline model

[Figure]

The (almost) ignorant model (Model-00) is a baseline model in the popular Nash–Sutcliffe model efficiency (NSE) criterion (e.g., Knoben et al., 2019, and SC1 therein for my comment).

As an alternative to it, I've suggested a simple(st) autoregressive model of order 2, AR(2):

$$Q(t) = Q(t-1) + [Q(t-1) - Q(t-2)] = 2Q(t-1) - Q(t-2), \tag{1}$$

This and their top–rated Model-07 (AR(3)) belong to the class of autoregressive processes. Their Equation (1) reads, omitting the subscript HOST to the discharge variable $Q$:

$$Q(t) = 0.0549 + 1.9266Q(t-1) - 1.2071Q(t-2) + 0.2685Q(t-3), \tag{2}$$

Similarity in terms of the coefficient between the two is striking. In an integer form, both are identical.

It would be instructive to score the performance of all the candidate models by the NSE criterion, both in its original and the newly suggested AR(2) form.

The AR(2)–based NSE will score Model-07 (AR(3)) again as a best performing model. It may differentiate more clearly the one–bucket (Model-02) and the two–buckets (Model-05) one. As expected, the latter performs better than the former (Lines 273-278), but the two are indistinguishable from each other on the authors' proposed model performance scale (Figure 3).

**2. Catchments as a quadratic reservoir**

The linear storage–discharge equation, $Q = S/K$ (Figure 1a) can be extended to a quadratic one below:

$$Q = (CS)^2, \tag{3}$$

In the absence of precipitation, $P(t) = 0$ for $\Delta t >> 1$ d, the recession hydrograph is linearized below:

$$-1/\sqrt{Q_t} = -1/\sqrt{Q_0} - C(t - t_0), \tag{4}$$

This has been called a negative inverse square root (NISR)–transformed recession flow model (Pelletier and Andreássian, 2020, and SC3 therein for my comment). This is in contrast to the universal logarithmic transformed one,

$$\log Q_t = \log Q_0 - (1/K)(t - t_0), \tag{5}$$

both having a single scale parameter $K$ or $C$.

**3. Training on transformed streamflow space**

As a consequence of data linearization described above, some prior transformation of the observed streamflow time series data may help reducing the model computational complexity (i.e. number of computing steps) as opposed to improving model performance (i.e. a consequence of applying hydrologic law, formulas, and equations)

(Lines 167-171).

As the case maybe, this can be the log or the NISR transformation of both a single reservoir (Figure 1a) and a two-parallel-reservoirs (Figure 1b) model, linear (Model-02 and 05) or quadratic as in Equation (3) above.

References

Knoben, W. J. M., Freer, J. E., and Woods, R. A.: Technical note: Inherent benchmark or not? Comparing Nash–Sutcliffe and Kling–Gupta efficiency scores, Hydrol.Earth Syst.Sci.,23,4323–4331,https://doi.org/10.5194/hess-23-4323-2019, 2019.

Pelletier, A. and Andréassian, V.: Hydrograph separation: an impartial parametrization for an imperfect method, Hydrol. Earth Syst. Sci.,24, 1171–1187, https://doi.org/10.5194/hess-24-1171-2020, 2020.

---

## Author Comment (AC1) · 17 Apr 2020

**Responses to short comment #1**

We thank John Ding for his comments on our manuscript. In the following, we reply to all comments one by one. Comments by John Ding are in blue.

I enjoy reading this Discussion paper from a hydrologic model performance perspective, and suggest the authors consider expanding the list of candidate models (Table 1) and their training methodology as follows:

1. A second–order autoregressive process as a baseline model

The (almost) ignorant model (Model-00) is a baseline model in the popular Nash–Sutcliffe model efficiency (NSE) criterion (e.g., Knoben et al., 2019, and SC1 therein for my comment).

As an alternative to it, I've suggested a simple(st) autoregressive model of order 2, AR(2):

$$Q(t) = Q(t-1) + [Q(t-1) - Q(t-2)] - 2Q(t-1) - Q(t-2) \qquad (1)$$

This and their top–rated Model-07 (AR(3)) belong to the class of autoregressive processes. Their Equation (1) reads, omitting the subscript HOST to the discharge variable Q:

$$Q(t) = 0.0549 + 1.9266Q(t-1) - 1.2071Q(t-2) + 0.2685Q(t-3) \qquad (2)$$

Similarity in terms of the coefficient between the two is striking. In an integer form, both are identical.

It would be instructive to score the performance of all the candidate models by the NSE criterion, both in its original and the newly suggested AR(2) form.

The AR(2)–based NSE will score Model-07 (AR(3)) again as a best performing model. It may differentiate more clearly the one–bucket (Model-02) and the two–buckets (Model-05) one. As expected, the latter performs better than the former (Lines 273-278), but the two are indistinguishable from each other on the authors' proposed model performance scale (Figure 3).

We thank John Ding for suggesting an alternative baseline model, and for suggesting alternative performance scores. We agree that adding more models, and NSE as an additional performance score, would be valuable if the focus of the paper would be about finding an optimal hydrological model for the Dornbinerach watershed. However, the main purpose of the manuscript is about introducing the bit-by-bit method. In this context, the reason behind applying a broad range of model types is to demonstrate the general applicability of the method. The performance of the models themselves is not a central element of the study. We think that giving more room to a discussion of model performance would distract from the key

message of the paper rather than strengthening it, therefore we prefer to keep the range of models and performance criteria as is.

Nevertheless, the questions raised by John Ding are interesting, and we are happy to provide some further details in the following:

AR-models

There is indeed strong similarity between the AR-3 model we used and the AR-2 model suggested by John Ding. We also fit an AR-2 model to our data by solving the Yule-Walker equations. The resulting model is Q(t) = 1.727 Q(t-1) – 0.743 Q(t-2) + 0.0751. Again, the coefficients are very similar to those suggested by John Ding, which suggests that the discharge time series the models were fit to are very similar in terms of their autoregressive properties.

Model training and model performance

Wherever applicable, we fit our models by minimizing mean absolute error (MAE). MAE is a popular performance score in hydrological modeling if good overall performance is sought. If good reproduction of high flow is important, NSE is the better alternative. MAE-optimization was done for model-02 and model-05. For model-00 and model-01, no calibration was required. Models -03, -04 and -06 apply the parameters of model-02. The AR-coefficients of model-07 were found by solving the Yule-Walker equations, and the coefficients of model-08 were found by minimizing the mean squared error, the standard loss function in the related Neural Network software package.

Table 1 contains the performance of all models used in the study in terms of Conditional Entropy (Hc), Nash-Sutcliffe efficiency (NSE) and Mean Absolute Error (MAE). We can see that in general, the model ranking is similar for the different performance criteria: Model-01 is the best for all criteria; model-02, model-03 and model-06 perform identical (as to be expected). Differences occur with respect to the worst performing model: According to Hc, it is model-00, for NSE and MAE it is model-04. Further, we can see that overall performance of the single-bucket model-02 and the two-bucket model-05 is not very good in terms of NSE, which indicates that the chosen model structure does not well reflect the complex hydrological behavior of the catchment (snow processes, occasional overland flow, seasonal patterns of evapotranspiration). However, the two-bucket model-05 outperforms the simpler single-bucket model-02 in terms of their calibration objective MAE , which is in accordance with expectations.

The bucket models were chosen for their simplicity, and clearly their performance leaves plenty of room for improvement. But as they just serve as a demonstration cases for the bit-by-bit method, we are convinced they are nevertheless useful and serve their purpose.

Table 1

| ID | type | Hc | NSE | MAE |
|---|---|---|---|---|
| Model-00 | Mean | 3.46 | 0 | 4.19 |
| Model-01 | Perfect | 0 | 1 | 0 |
| Model-02 | 1 bucket | 2.89 | 0.08 | 4.78 |
| Model-03 | 02+dt 1 min | 2.89 | 0.08 | 4.78 |
| Model-04 | 02+integer | 3.23 | -1.8 | 7.99 |
| Model-05 | 2 bucket | 2.85 | -0.11 | 4.54 |
| Model-06 | 02+iterative | 2.89 | 0.08 | 4.78 |
| Model-07 | AR-3 | 0.66 | 0.99 | 0.18 |
| Model-08 | ANN | 3.37 | 0.12 | 3.50 |

**2. Catchments as a quadratic reservoir**

The linear storage–discharge equation, Q = S/K (Figure 1a) can be extended to a quadratic one below:

$$Q = (CS)^2 \tag{3}$$

In the absence of precipitation, P(t) = 0 for Δt >>1 d, the recession hydrograph is linearized below:

$$\frac{-1}{\sqrt{Q_t}} = \frac{-1}{\sqrt{Q_0}} - C(t - t_0) \tag{4}$$

This has been called a negative inverse square root (NISR)–transformed recession flow model (Pelletier and Andreássian, 2020, and SC3 therein for my comment). This is in contrast to the universal logarithmic transformed one,

$$logQ_t = logQ_0 - \left(\frac{1}{K}\right)(t - t_0) \tag{5}$$

both having a single scale parameter K or C.

We agree that extending the range of reservoir candidates from linear to higher-order relations adds flexibility to models and potentially allows better calibration to a given catchment. However, the main purpose of the manuscript is about introducing the bit-by-bit method. In this context, the reason behind applying a broad range of model types is to demonstrate the general applicability of the method, and the performance of the models themselves is not a central element of the study. We therefore prefer to keep the range of models as is (please also see our reply to comment 1).

3. Training on transformed streamflow space

As a consequence of data linearization described above, some prior transformation of the observed streamflow time series data may help reducing the model computational complexity (i.e. number of computing steps) as opposed to improving model performance (i.e. a consequence of applying hydrologic law, formulas, and equations) (Lines 167-171).

As the case maybe, this can be the log or the NISR transformation of both a single reservoir (Figure 1a) and a two-parallel-reservoirs (Figure 1b) model, linear (Model-02 and 05) or quadratic as in Equation (3) above.

We agree that the suggestions by John Ding can help to better fit the models to the given data. But for the reasons already given in the replies to the previous two comments, we prefer to keep the models as they are.

Yours sincerely,

Uwe Ehret, on behalf of all co-authors

---

## Short Comment (SC2) · 21 Apr 2020

Dear Azmi et al.,

I very much enjoyed reading your submitted manuscript. However, I would like add two notes on this interesting discussion:

Firstly, I was missing from the manuscript that we not only use models to distill that one perfect equation but continuously use them to further our system understanding. As you noted quite precisely natural systems are very complex and modeling, for example a watershed, requires to take into account many variables and simulate a large

system. Because of that, and also because these systems almost always contain human interactions which additionally make everything more complex, building a model is never a finished process that ends with one equation that best describes the system. Often it is a first "educated guess" that is then used as a foundation to understand the system further e.g. by using sensitivity analysis. It would be great if this is a little more reflected in this paper. I would also like to mention Wagener, T., McIntyre, N., Lees, M.J., Wheater, H.S. and Gupta, H.V. (2003), Towards reduced uncertainty in conceptual rainfall‐runoff modelling: dynamic identifiability analysis. Hydrol. Process., 17: 455-476. doi:10.1002/hyp.1135 as a possible citation.

A second discussion point I would like to raise is that in line 308 you clearly state that you are maintaining a information-theoretic point of view, which is good and clearly sets the scope of the discussion; nevertheless, I think an important point is missing: the skill of a researcher to implement a model well enough. Lets say, for the sake of the argument, that our perceptual model (a term coined by Keith Beven) of reality is almost perfect and with our modeling approach, whatever technique we apply (bucket, neural network ...), we would theoretically reach a high level of model performance. But because implementing models is a hugely difficult task, amplified by the lack of computer science and computer engineering background in the natural sciences (Hutton, Christopher, et al. "Most computational hydrology is not reproducible, so is it really science?." Water Resources Research 52.10 (2016): 7548-7555.), we may reach a very high computational complexity but possibly also a low model performance. I think this discussion should be reflected in your paper. It doesn't make your approach less applicable but highlights that looking only at this metric is not enough to guide the community to better research!

Small notes on the abstract: 16: "length of the model" it is explained later in the manuscript but very misleading here. I was thinking of lines of code or runtime when reading it first 29: "low performance" unclear if it refers to computational performance or model fit to observations or expected system behavior

With regards, Robert Reinecke.

---

## Author Comment (AC2) · 23 Apr 2020

**Responses to short comment #2**

We thank Robert Reinecke for his interest in our manuscript and for his comments! In the following, we reply to all comments one by one. Comments by Robert Reinecke are in blue.

I very much enjoyed reading your submitted manuscript. However, I would like add two notes on this interesting discussion:

Firstly, I was missing from the manuscript that we not only use models to distill that one perfect equation but continuously use them to further our system understanding. As you noted quite precisely natural systems are very complex and modeling, for example a watershed, requires to take into account many variables and simulate a large system. Because of that, and also because these systems almost always contain human interactions which additionally make everything more complex, building a model is never a finished process that ends with one equation that best describes the system. Often it is a first "educated guess" that is then used as a foundation to understand the system further e.g. by using sensitivity analysis. It would be great if this is a little more reflected in this paper. I would also like to mention Wagener, T., McIntyre, N., Lees, M.J., Wheater, H.S. and Gupta, H.V. (2003), Towards reduced uncertainty in conceptual rainfall runoff modelling: dynamic identifiability analysis. Hydrol. Process., 17: 455-476. doi:10.1002/hyp.1135 as a possible citation.

Agreed. Model building, and the scientific endeavor in general is mostly incremental rather than an one-off process with a final absolute outcome. Moreover, we are also aware that "in the context of system investigation, models can also be seen as laboratories, designed and deployed by humans to investigate to a given extent and under given conditions aspects of a presumptive phenomenological reality" (quote from: Perdigão, 2017). In that sense, we agree with Robert Reinecke that what we suggest in the manuscript is a tool "which can be used together to guide model analysis and optimization in a pareto trade-off manner" (see p 13 line 326-327), and we will add to a revised version of the manuscript a phrase that his happens in the general setting of incremental learning.

A second discussion point I would like to raise is that in line 308 you clearly state that you are maintaining a information-theoretic point of view, which is good and clearly sets the scope of the discussion; nevertheless, I think an important point is missing: the skill of a researcher to implement a model well enough. Let's say, for the sake of the argument, that our perceptual model (a term coined by Keith Beven) of reality is almost perfect and with our modeling approach, whatever technique we apply (bucket, neural network ...), we would theoretically reach a high level of model performance. But because implementing models is a hugely difficult task, amplified by the lack of computer science and computer engineering background in the

natural sciences (Hutton, Christopher, et al. "Most computational hydrology is not reproducible, so is it really science?." Water Resources Research 52.10 (2016): 7548-7555.), we may reach a very high computational complexity but possibly also a low model performance. I think this discussion should be reflected in your paper. It doesn't make your approach less applicable but highlights that looking only at this metric is not enough to guide the community to better research!

We fully agree with Robert Reinecke that a perfect conceptual model is just a necessary, but not a sufficient precondition for perfect predictions, and that the actual coding of a model contains endless opportunities to mess things up. Along the lines of Robert Reinecke's comment, we can state that poor coding will increase model computational complexity (e.g. think of redundant loops that can be replaced by a one-go matrix computation, or overly fine-grained time-stepping), and wrong coding will reduce model performance (or increase information loss). We suggest adding a phrase to the summary and conclusion at the place mentioned in the previous comment, stating that bit-by-bit is also a tool to promote better (less poor and less wrong) coding, together with a reference to the Hutton et al. (2016) paper.

Small notes on the abstract: 16: "length of the model" it is explained later in the manuscript but very misleading here. I was thinking of lines of code or runtime when reading it first

Agreed. We suggest adding the following explanation:

"The basic dimensions of computer model parsimony are descriptive complexity, i.e. the size  of the model itself, which can be measured by the disk space it occupies, and computational complexity, i.e. the model's effort to provide output.

29: "low performance" unclear if it refers to computational performance or model fit to observations or expected system behavior.

Agreed. As "performance" already appears in line 14, we suggest explaining it there:

"Measuring performance and parsimony for computer models is therefore a key theoretical and practical challenge for 21st century science. "Performance" here refers to a model's ability to reduce predictive uncertainty about an object of interest. The basic dimensions ..."

Yours sincerely,

Uwe Ehret, on behalf of all co-authors

**References**

Perdigão, R.A.P. (2017). Fluid dynamical systems: From quantum gravitation to thermodynamic cosmology. M-DSC Monograph. (Hard copies for order at www.fluiddynamicalsystems.com).

---

## Short Comment (SC3) · 25 Apr 2020

**On new Table 1**

I appreciate the thoughtful response by the authors. The new Table 1 is very informative, as it scores the candidate models by the new Hc (conditional entropy) criterion, as well as two additional ones: the NSE (Nash-Sutcliffe efficiency) and MAE (maximum absolute error).

[Figure]

In my view, the "Perfect" model (model-01) is an AR-0 (an autoregressive model of order 0), a null model, i.e. observed data is the model.

Thanks for the opportunity to offer my initial comment (SC1) and this follow-up.

―――――――――――――――

---

## Short Comment (SC4) · 25 Apr 2020

**Correction to the term MAE**

MAE is not the **Maximum** Absolute Error as written by me in SC3. It is the **Mean** Absolute Error as used by the authors in AC1.

---

## Author Comment (AC3) · 26 Apr 2020

Dear Editor, dear John Ding,

We are glad that John Ding finds our replies to his comments useful. Furthermore, we agree that our model-01 could be called a zero-order autoregressive model, with Q(t) as the predictor for Q(t).

Yours sincerely, Uwe Ehret, on behalf of all co-authors

---

## Author Comment (AC4) · 26 Apr 2020

Dear Editor, dear John Ding,

Indeed MAE is Mean Absolute Error.

Yours sincerely, Uwe Ehret, on behalf of all co-authors
* * *

---

## Referee Comment (RC1) · Elena Toth (Referee) · 29 May 2020

The Technical Note addresses a very timely research question, covered also in a recent WRR debate on the role of Information Theory for helping to understand the complexity of Earth Systems. In particular, the Note attempts to provide some examples for testing/sustaining some of the ideas presented in the WRR debate, that includes also one contribution by two of the Authors (Weijs and Ruddell, 2020). I find the theme extremely interesting and I believe that proposing a way (using 'Strace') to calculate computational effort without the need to refer to the specific machine where the computations run is indeed brilliant and novel and worthy to be published. On the other

hand, I confess that on one hand I find the Note a bit too theoretical and on the other hand I have some doubts on the soundness of the comparison that is presented.

The note often refers to the Weijs and Ruddell (2020) paper but without well clarifying the relationship between such paper and the present work (what is the content of the previous paper and what is different here). A good part of the theoretical discussion, indeed a truly philosophical one, as such was the 'line' of the WRR debate, is repeated here, in a first part (three pages of introduction) that is definitely too long and too much theoretical for a technical note and may be substantially shortened, referring to the previous publication.

But my main concern is that the comparison of the models is not fair, since they do not make use of the same information and this is instead crucial in a work focussing on information theory. Looking at Table 1, last column, we may see that the data used for running the models are not the same. In fact, the bucket models (Models 02 and 05) do not use any streamflow data in any way for the simulation but only for calibrating the parameters. The same holds for the ANN (Model-08) since only P is provided in input. On the other hand, the autoregressive model (Model-07) uses only past streamflow values as input. It is well known that for a short lead time (the models are here used as simulation models, with lead-time equal to one), the recent measures of the streamflow (Q) is much more informative than the rainfall values, that in real-time flow forecasting become more and more important when the lead-time increases, since Q encapsulates a lot of useful information on the catchment behaviour, and it may be seen as a very good approximation of the catchment conditions. Therefore it was easily predictable that the autoregressive model (Model-07) would have outperformed the other models, independently of its complexity, due to the different input information they use.

Thus, leaving aside the analysis of the performances (expected, due to the setting up of the models), the interesting part of the results is the analysis of the complexity. Section 3.2 and Figure 3 show that, a part from Model-03 and Model-08 (ANN), all the other complexities are very close. The reason for the high computational complexity

of Model-03 is the excessively (and not necessary) fine time step. The reason for the high computational complexity of Model-08, that is the Artificial Neural Network, may certainly be inherent in the structure of these kind of models, that tend to have a relatively high number of parameters (but the internal parameters are in some cases not all influential and since, despite the high number of parameters, ANNs generally work very well on independent data, they cannot be blamed of overfitting/overtraining). But in addition, in this case the ANN model is not only fed by the "wrong" input (P instead of Q), but its architecture is also certainly more complex than needed: why using 10 hidden nodes? If it were used, as it should, in a way that is consistent with the regressive model, it should be fed by the last streamflow values rather than (or in addition to) some past rainfall (needed especially if considering longer lead-times) and it would perform much better than now. And probably a few hidden nodes would be more than enough (as proved in many previous works where such models rival with more complex conceptual models in forecasting/updating mode), so its complexity would be less.

Due to the potential of the bit-by-bit concept, and the utility to be able to measure computational complexity through 'Strace', I do encourage the Authors to perform and present a more fair comparison and then focussing and explaining the differences, in performances and complexity, found in models that use indeed the same input information content (and have the most parsimonious structure that is possible).

SPECIFIC COMMENTS

Abstract: ll.12-20: may be summarised.

Pages 2 to 4 may be summarised in one page, referring to Wejis and Ruddell (2020) for the philosophical discussion.

Eq. 1: I would suggest to move eq 1 inside Table 1 (Model-07 row)

ll. 155-158: actually I would have found very interesting an evaluation of out-of-sample

performance of the models, since this is indeed crucial for data-driven models and it would be very useful to understand what each model is able to infer on the behaviour of the basin on independent data, to analyse their generalisation ability.

Second part of Section 2.4.1: I think that more detail on the meaning and computation of entropy is necessary, since it is a 'niche' not widely known to the readers.

ll. 266-269: can you explain the differences in computational complexity between Models 00 and 01? I would have expected their complexity to be practically null for both, since they do not need to make computations at each step. . .

---

## Referee Comment (RC2) · Anonymous Referee #2 · 12 Jun 2020

Summary: As the title suggests, the authors present a practical approach to evaluate model computational complexity vs. model performance.

Comments: this is suitable technical note. I however wouldnot so easily agree with the authors' claim that their approach is a general approach if they mean in terms of its theoretical underpinning. General perhaps in terms of how easily it can be applied. Nonethess this is an interesting article that readers can learn from and apply the methodology in diverse settings. Following are my comments in detail.

- How max parsimony + max performace -> max generalizability? Theoretical rigor behind the claim in missing

[Figure]

- Concept of information loss assumes full specification of the data generating process, which often is not the case in hydrological modeling. Please elaborate further how this is dealt with

- In this context thinking of models as compression algorithms of data is shallow in its treatment of what complexity means in terms of learning from patterns, especially when patterns are generated from complex data generating process. I can understand the concept being a good one in describing model complexity appropriately when we know the data generating process and are playing with its approximations and trading that off with information loss incurred by the approximations. So the authors claim of universality is overdone in real world hydrological systems, perhaps it may work in Shannon's communication systems.

- Related to the above, it is for this reason that synthetic cases may be easier to demonstate. Author's claim to universality should first provide a rigorous theoretical treament that has not even been provided in the WRR paper that the authors allude to.

- That is the reason why the authors attempt to extend it to real world case studies is not constructive unless the error model of the residuals is completely specified (or known).

- I am not at all clear how computational complexity is linked to inference. This is where the paper lost me in its attempt to connect this paper to their earlier WRR paper. Here while authors talk about inference without reference to predictive performance, no clear theory on how computational complexity is linked to generalizability is given.

- Even if 'generalization' laws have been found, how good they are depend on how well they hold on unseen data, i.e. predictive uncertainty

- I was totally lost in the philosophical arguments at the end of the introduction paper. Please delete, it appears to have been placed to impress the reader. I am reacting to it in quite an opposite manner

- the way Prob for entropy measure has been calculated is in itself a model that depends on the choice and number of bins. That has implications for how well Prob has been estimated from limited data in terms of how such frequency estimates converge to true Prob (ie it has its own complexity challengs) that the approach so very much relies on. Perhaps this can be discussed in bit more detail.

Finally two major comments:

- the authors should show predictive performance to demonstrate generalizability. Or validation, even if in narrative form, by comparing their conclusions with what other authors, not linked to information theory applied to water, have said.

- the authors again need to place their finding in the landscape of other complexity studies, especially in modeling MOPEX catchments, in hydrology. How do their conclusions regarding complexity compare with the narrative presented here? This will only add value to an already large literature set of hydrological model complexity, esp wrt to streamflow..

---

## Author Comment (AC5) · 9 Jul 2020

**Responses to comments by referee #1 (Elena Toth)**

Dear Editor, dear Elena Toth,

We thank the first referee, Elena Toth, for the constructive review of our manuscript and the detailed comments, which will help us to sharpen our arguments. We will in the following reply to the comments point by point. The Referee comments are in blue.

Comment 1: The Technical Note addresses a very timely research question, covered also in a recent WRR debate on the role of Information Theory for helping to understand the complexity of Earth Systems. In particular, the Note attempts to provide some examples for testing/sustaining some of the ideas presented in the WRR debate, that includes also one contribution by two of the Authors (Weijs and Ruddell, 2020). I find the theme extremely interesting and I believe that proposing a way (using 'Strace') to calculate computational effort without the need to refer to the specific machine where the computations run is indeed brilliant and novel and worthy to be published. On the other hand, I confess that on one hand I find the Note a bit too theoretical and on the other hand I have some doubts on the soundness of the comparison that is presented. The note often refers to the Weijs and Ruddell (2020) paper but without well clarifying the relationship between such paper and the present work (what is the content of the previous paper and what is different here). A good part of the theoretical discussion, indeed a truly philosophical one, as such was the 'line' of the WRR debate, is repeated here, in a first part (three pages of introduction) that is definitely too long and too much theoretical for a technical note and may be substantially shortened, referring to the previous publication.

Reply 1: Thank you for this comment, which made us aware that we have to i) maker better clear the contribution of the paper, and ii) explain its connection to Weijs and Ruddell (2020).

The contribution of the paper is twofold: Firstly we make a practical suggestion on how to measure computational effort of computer-based models. Secondly, we embed this suggestion into the larger and more theoretical topic of model and theory building and related guiding principles such as Occam's Razor or the approach suggested by Weijs and Ruddell (2020). Apparently we haven't clearly explained the relation of the 'bit by bit' concept and the other concepts, and we also have not included a proper description of the classical 'validation set' approach and its relation to the other approaches. We suggest adding a figure and related discussion to the manuscript (see below) for clarification.

[Figure]

(a) Aspects of model and theory evaluation

(b) Guidelines for model and theory development

In short, we will discuss:

- Weijs and Ruddell (2020) extend Occam's Razor – which evaluates descriptive complexity only – by additionally measuring performance. Also, they measure both criteria in a unit [bit] and using a unified mathematics which allows a joint treatment and comparison between the two, including their inherent tension and complementarity.
- Classical validation set approaches seek to maximize model performance on unseen data. By using unseen data, general/parsimonious model approaches are favored, and overfitting is avoided. The approach by Weijs and Ruddell (2020) assures generality/parsimony by including descriptive complexity into their maximization function. For the latter we will include references to relevant literature from Algorithmic Information Theory.
- The bit by bit approach adds another key dimension of model complexity: computational complexity, measured by 'strace' in unit [bit]. It is important to note that computational complexity is neither a surrogate for measuring performance, nor for measuring descriptive complexity. In order to provide a comprehensive evaluation of a model, computational complexity can be either combined with the approach by Weijs and Ruddell (2020), OR with a validation set approach. In both cases, there is a guard for overfitting (descriptive complexity in the first, evaluation on unseen data in the second). In the current version of the

manuscript, we combined computational complexity with performance on a data set seen through calibration - as in Weijs and Ruddell (2020) - but did not add a descriptive complexity control. This is inconsistent with the above explanation. We therefore suggest that in a revised version of the paper, we will use computational complexity in combination with performance measured by information loss in a validation set approach. This is in accordance with the referee's suggestion, and readers from the hydrological community will find this easier to relate to as it follows established procedures of cal/val.

- We will further describe an approach - and suggest it for future research - where computational complexity is combined with the approach by Weijs and Ruddell (2020). This avoids the need for splitting available data into cal and val subsets, but still favors general/parsimonious models. In addition, as all criteria can be measured in unit [bit], this allows a joint treatment and comparison of all key aspects of model evaluation.

We hope that by the figure and the explanation, the relation between the different guiding principles, and their relation to the scientific process of model/theory building/evaluation becomes more clear. We suggest keeping the existing in-depth introduction to the theory, as placing the bit by bit approach into this framework is one main goal of the paper.

Comment 2: But my main concern is that the comparison of the models is not fair, since they do not make use of the same information and this is instead crucial in a work focussing on information theory. Looking at Table 1, last column, we may see that the data used for running the models are not the same. In fact, the bucket models (Models 02 and 05) do not use any streamflow data in any way for the simulation but only for calibrating the parameters. The same holds for the ANN (Model-08) since only P is provided in input. On the other hand, the autoregressive model (Model-07) uses only past streamflow values as input. It is well known that for a short lead time (the models are here used as simulation models, with lead-time equal to one), the recent measures of the streamflow (Q) is much more informative than the rainfall values, that in real-time flow forecasting become more and more important when the lead-time increases, since Q encapsulates a lot of useful information on the catchment behaviour, and it may be seen as a very good approximation of the catchment conditions. Therefore it was easily predictable that the autoregressive model (Model-07) would have outperformed the other models, independently of its complexity, due to the different input information they use. Thus, leaving aside the analysis of the performances (expected, due to the setting up of the models), the interesting part of the results is the analysis of the complexity. Section 3.2 and Figure 3 show that, a part from Model-03 and Model-08 (ANN), all the other complexities are very close. The reason for the high computational complexity of Model-03 is the excessively (and not necessary) fine time step. The reason for the high computational complexity of Model-08, that is the Artificial Neural Network, may certainly be inherent in the structure of these kind of models, that tend to have a relatively high number of parameters (but the internal parameters are in some cases not all influential and since, despite the high number of parameters, ANNs generally

work very well on independent data, they cannot be blamed of overfitting/overtraining). But in addition, in this case the ANN model is not only fed by the "wrong" input (P instead of Q), but its architecture is also certainly more complex than needed: why using 10 hidden nodes? If it were used, as it should, in a way that is consistent with the regressive model, it should be fed by the last streamflow values rather than (or in addition to) some past rainfall (needed especially if considering longer lead-times) and it would perform much better than now. And probably a few hidden nodes would be more than enough (as proved in many previous works where such models rival with more complex conceptual models in forecasting/updating mode), so its complexity would be less. Due to the potential of the bit-by-bit concept, and the utility to be able to measure computational complexity through 'Strace', I do encourage the Authors to perform and present a more fair comparison and then focussing and explaining the differences, in performances and complexity, found in models that use indeed the same input information content (and have the most parsimonious structure that is possible).

Reply 2: Thank you for this comment, which indicates that we need to better explain the take-home messages from the model comparison. The two key messages are that

- computational complexity as measured by 'strace' is sensitive to ALL computational aspects of a model: the amount of required forcing data, the size of the model itself, its time stepping, spatial resolution, numerical scheme, etc. ,
- 'strace' can be applied to ANY computer-based model, i.e. it is general within its range of application (evaluation of computer-based models).

The main goal of the model comparison is therefore to demonstrate is applicability across a range of different model concepts (therefore we included bucket models, autoregressive models, ANN models), and its sensitivity to different computational aspects (therefore we included variations of a bucket model in terms of time stepping). It is therefore neither required nor useful to establish models that all rely on the same input data, because the bit by bit approach might be used for exactly the purpose of comparing two models relying on different input data. In the light of the two key take-home messages mentioned above, we hope it also becomes clear that the actual performance of the models, and their performance comparison is not central to this paper, and that they might even distract from the main messages. Nevertheless we see the referee's point that the ANN used in the study was set up in a far-from-optimal manner, which obviously appears as an unfair treatment compared to the other models.

We therefore suggest to include into a revised version of the manuscript:

- A better explanation of the purpose and key messages of the model comparison
- Replacing the current ANN by a more appropriate one with less nodes

- As explained in reply 1, we will calibrate the models in a calibration subset of the data, and present results for a validation subset.

Specific comments

Comment 3: Abstract: ll.12-20: may be summarised.

Reply 3: Respectfully we disagree. A brief introduction of the main terms in the abstract helps to put the 'bit by bit' concept into perspective, and helps conveying the two main goals of the manuscript (see our reply to comment 1). We will however re-read and potentially re-write the abstract in the light of the added figure and changed experimental design (see our reply to comment 1).

Comment 4: Pages 2 to 4 may be summarised in one page, referring to Wejis and Ruddell (2020) for the philosophical discussion.

Reply 4: Please see our reply to comment 3.

Comment 5: Eq. 1: I would suggest to move eq 1 inside Table 1 (Model-07 row)

Reply 5: Agreed. We will do so in a revised version of the manuscript.

Comment 6: ll. 155-158: actually I would have found very interesting an evaluation of out-of-sample performance of the models, since this is indeed crucial for data-driven models and it would be very useful to understand what each model is able to infer on the behavior of the basin on independent data, to analyse their generalisation ability.

Reply 6: In a revised version of the manuscript, we will fit the models in a calibration period, and present results for a validation period. Please see also our related reply to comment 1.

Comment 7: Second part of Section 2.4.1: I think that more detail on the meaning and computation of entropy is necessary, since it is a 'niche' not widely known to the readers.

Reply 7: Agreed. We will add the entropy equation, a brief explanation, and references to relevant literature to a revised version of the manuscript.

Comment 8: ll. 266-269: can you explain the differences in computational complexity between Models 00 and 01? I would have expected their complexity to be practically null for both, since they do not need to make computations at each step. . .

Reply 8: The computational complexity of these model arises from preparing the output, and actually writing the output. The difference between the two is that for the first, a single number is written into the output array, for the second the observed timeseries is read, and then written into the output array. The related Matlab code is:

| Model_00 | Model_01 |
|---|---|
| ```% assign the mean flow
q_host_mean = 4.5998;

% make the prediction for each time step
output_00 = zeros(87650,1) + q_host_mean;``` | ```% load the observed Q data
load q_host

% copy the input ('q_host') to the output ('output')
output_01 = q_host;``` |

So the effort is really low, but nevertheless an array of double-precision numbers needs to be copied. Interestingly, as the results show, the related effort is higher than for model_04, even if the latter has more elaborate code. The reason is that the latter acts on integer-precision variables.

Code of model_04:

```
% load the input data
load p_ebni_int

% get parameters
len = length(p_ebni_int); % length of the data set

% hydrological model setup
K = 55;                        % retention constant = mean transit time [h]
qsim = int8(zeros(len,1));     % reservoir discharge [mm/h]
S = int8(0);                   % initialize the reservoir fill level [mm]

% loop over time
for t = 2 : len
    S = S + p_ebni_int(t);         % storage change due to rainfall input
    qsim(t) = S / K;   % discharge as f(storage volume)
    S = S - qsim(t);               % storage change due to discharge
end

% convert the discharge from [mm/h] into [m³/s]
output_04 = qsim * 31.8888888;
```

Yours sincerely,

Uwe Ehret, on behalf of all co-authors

---

## Author Comment (AC6) · 9 Jul 2020

**Responses to short comments by referee #2**

Dear Editor, dear Referee,

We thank the second referee for the review of our manuscript and the detailed comments. We will in the following reply to the comments point by point. The Referee comments are in blue.

Comment 1: Summary: As the title suggests, the authors present a practical approach to evaluate model computational complexity vs. model performance.

Comment 2: Comments: this is suitable technical note. I however would not so easily agree with the authors' claim that their approach is a general approach if they mean in terms of its theoretical underpinning. General perhaps in terms of how easily it can be applied. Nonetheless this is an interesting article that readers can learn from and apply the methodology in diverse settings. Following are my comments in detail.

Reply 2: Thank you for this comment, which made us aware that we have to better explain the meaning of 'general' in our manuscript. It is used in two different but related contexts: The first refers to the use of 'strace' as a tool to measure computational complexity of a model. It is general in the sense that i) within its domain of application (computer-based models), it can be applied to all models, whether or not the source code is available, and ii) it is sensitive to all computational aspects of a model (the amount of required forcing data, the size of the model itself, its time stepping, spatial resolution, numerical scheme, etc.).

The second use refers to the 'bit by bit' concept: It is general within its domain of application (evaluation of models in the process of science) in the sense that i) it covers all key aspects (performance and complexity), and ii) that by expressing all aspects in unit [bit], it allows their combination and comparison. This would not be possible if e.g. performance were expressed as RMSE, and computational complexity in [s].

In a revised version of the manuscript, we will better explain the meaning of 'general'.

Comment 3: How max parsimony + max performance -> max generalizability? Theoretical rigor behind the claim is missing

Reply 3: This claim refers to the idea that when describing data well with a simple model, it tends to work well for prediction of unseen data. This is building on Occam's razor. The theoretical basis of this claim is addressed in detail in Weijs and Ruddell (2020) and references therein. This is widely accepted in practice and, for example, visible in the Akaike Information Criterion (a performance term + a complexity penalization). Algorithmic information theory provides a deeper theoretical basis, and explains that shorter descriptions are more likely to be

generalizable (predictive). The theoretical rigor draws upon many concepts beyond the typical expertise of the intended audience (universality of computation, self-delimiting Turing machines etc.), that we do not think are helpful to discuss here as it is not the main focus of the paper. Hence we refer to the WRR paper that in turn refers to the AIT literature, such as the convergence results by Solomonoff (1978).

On an intuitive level, the idea is that we can see physical processes as performing computations, simple computations are more likely to arise naturally, and thus have a higher a priori probability (see algorithmic probability). This is a quantification of Occam's razor and expresses a belief of structure in the universe. Occam's razor is an essential part of finding descriptions that generalize beyond the given data set, as without it there are infinitely many equally valid explanations. The explanations that are simple tend to apply more easily to new situations and thus are more generalizable. This is the reason why inductive inference works at all, and why intelligent beings tend to have an evolutionary advantage in this universe.

One more note about theoretical rigor: Even with AIT, we cannot prove that inductive inference works, but we can better formally describe and measure how well it works in practice (so the only evidence that induction works is itself an inductive inference).

Comment 4: Concept of information loss assumes full specification of the data generating process, which often is not the case in hydrological modeling. Please elaborate further how this is dealt with

Reply 4: Thank you for this comment, which shows that we need to better explain the usage of 'information loss'. We have two replies:

- The referee correctly states that the process generating the observations we have is typically not fully known in earth science problems, except for virtual reality settings. This is the case no matter whether we measure performance by RMSE, NSE, or information loss. So the best we can do is to measure performance against the observations we have, and hope that it allows conclusions that are also valid for the true underlying data generating system. In that sense, we can use the distribution and dependencies of the data as a benchmark, against which information losses can be calculated.
- Alternatively, if we want to emphasize the limited information contained in the data about the system (e.g. in cases of very few observations), we can start from the opposite end: We can specify maximum entropy (=minimum prior knowledge) distributions for all system variables of interest, and then measure the information gain of the available data against this flat prior. In such a case, the y-axis in figure 3 shows information gain, but the overall interpretation remains the same.

Weijs and Ruddell (2020), which we refer to throughout the text, use information losses because they directly translate to a description length. For reasons of comparability we prefer

to stick to the same measure in the manuscript. This has the advantage that all measures are negatively oriented scores, where lower is better. We will add a related explanation to a revised version of the manuscript.

Comment 5: In this context thinking of models as compression algorithms of data is shallow in its treatment of what complexity means in terms of learning from patterns, especially when patterns are generated from complex data generating process. I can understand the concept being a good one in describing model complexity appropriately when we know the data generating process and are playing with its approximations and trading that off with information loss incurred by the approximations. So the authors claim of universality is overdone in real world hydrological systems, perhaps it may work in Shannon's communication systems.

Reply 5: We would like to reply to this comment in two parts: We think algorithmic information theory is far from shallow, but we do agree there are still challenges to be overcome for its practical application in hydrology.

Since descriptive complexity is not the main focus of this paper, we did not go into much detail, but more discussion on this can be found in the WRR paper on Occam's Razor. One thing to note is that however complex a hydrological system may be, all its processes can – in principle – be simulated by a Universal Turing Machine. There are connections at the deepest level between physics, information, and computation (See e.g. the Church-Turing-Deutsch thesis, black hole thermodynamics, the Landauer's principle), therefore AIT certainly has bearing on inference outside the world of computer science and communication systems. Whatever real world data set is produced by whatever generating process, it will have a finite Kolmogorov complexity, and in principle a binary computer program could perfectly reproduce it. Approaching that Kolmogorov complexity with finding shortest descriptions of all dependencies in the data is exactly the same process as modeling the patterns in that data. Note that there is no reference to an underlying data generating mechanism here, only to the data itself, which in the end is our only window on reality. Now of course our hope is that the model, or compression algorithm, found has some functional similarity with the data generating process, and we can use it for predictions. And that usually works, because the universe typically is ruled by a decent bit of order and simplicity (with the exception of my desk).

Now as for practical application, even before theoretical computability issues come in, practical limitations will make it impossible to pursue finding the full likeliest algorithm that produced the data. Especially since the typical data sets used in hydrology do not fully specify the system, or even do not specify everything we know about it. However, there are several ways in which this lack of completeness can be dealt with by describing the variations unexplained by the patterns we found literally (which is the equivalent of viewing the data generating process as stochastic).

Comment 6: Related to the above, it is for this reason that synthetic cases may be easier to demonstrate. Author's claim to universality should first provide a rigorous theoretical treatment that has not even been provided in the WRR paper that the authors allude to.

Reply 6: Respectfully, we do not understand the referee's point here: In the manuscript we provide applications of typical hydrological models to real-world data. However, we understood that we should use the term 'universality' more carefully. In a revised version of the manuscript, we will replace 'universality' with 'generality', and better explain the use of the latter (see our reply to comment 2).

Comment 7: That is the reason why the authors attempt to extend it to real world case studies is not constructive unless the error model of the residuals is completely specified (or known).

Reply 7: We are not sure we understand the referee's concern here. We assume that the mentioned error model of the residuals specifies the disagreement between the data generating system and the observations thereof. If this is the case, please see our reply to comment 4. If this is not the case, please explain.

Comment 8: I am not at all clear how computational complexity is linked to inference. This is where the paper lost me in its attempt to connect this paper to their earlier WRR paper. Here while authors talk about inference without reference to predictive performance, no clear theory on how computational complexity is linked to generalizability is given.

Reply 8: Thank you for this comment. It shows that we need to better explain the relation among the concepts introduced or discussed in the manuscript. In order to increase overall clarity, we suggest adding a figure and related discussion to the manuscript (see below).

[Figure]

(a) Aspects of model and theory evaluation

(b) Guidelines for model and theory development

In short, we will discuss:

- Weijs and Ruddell (2020) extend Occam's Razor – which evaluates descriptive complexity only – by additionally measuring performance. Also, they measure both criteria in unit [bit], which allows a joint treatment and comparison.
- Classical validation set approaches seek to maximize model performance on unseen data. By using unseen data, general/parsimonious model approaches are favored, and overfitting is avoided. The approach by Weijs and Ruddell (2020) assures generality/parsimony by including descriptive complexity into their maximization function. For the latter we will include references to relevant literature from Algorithmic Information Theory.
- The bit by bit approach adds another key dimension of model complexity: computational complexity, measured by 'strace' in unit [bit]. It is important to note that computational complexity is neither a surrogate for measuring performance, nor for measuring descriptive complexity. In order to provide a comprehensive evaluation of a model, computational complexity can be either combined with the approach by Weijs and Ruddell (2020), OR with a validation set approach. In both cases, there is a guard for overfitting (descriptive complexity in the first, evaluation on unseen data in the second). In the current version of the manuscript, we combined computational complexity with performance on a data set seen

through calibration - as in Weijs and Ruddell (2020) - but did not add a descriptive complexity control. This is inconsistent with the above explanation. We therefore suggest that in a revised version of the paper, we will use computational complexity in combination with performance measured by information loss in a validation set approach. We hope that readers from the hydrological community will find this easy to relate to as it follows established procedures of cal/val.

- We will further describe an approach - and suggest it for future research - where computational complexity is combined with the approach by Weijs and Ruddell (2020). This avoids the need for splitting available data into cal and val subsets, but still favors general/parsimonious models. In addition, as all criteria can be measured in unit [bit], this allows a joint treatment and comparison of all key aspects of model evaluation.

We hope that by the figure and the explanation, the relation between the different guiding principles, and their relation to the scientific process of model/theory building/evaluation becomes more clear. We suggest keeping the existing in-depth introduction to the theory, as placing the bit by bit approach into this framework is one main goal of the paper.

A direct reply to the referee comment: Computational complexity is not directly linked to inference, rather it provides another key facet of model evaluation that can be used in addition to the criteria used by Weijs and Ruddell (2020). Also, we do not claim that computational complexity provides guidance on the generality of models or laws, our claim is rather that measuring computational complexity by 'strace' and the 'bit by bit' approach are generally applicable (please see also our replies to comment 2).

Comment 9: Even if 'generalization' laws have been found, how good they are depend on how well they hold on unseen data, i.e. predictive uncertainty

Reply 9: Please see our reply to comment 8 about how the different concepts discussed in the manuscript (Occam's razor, Weijs and Ruddell 2020, Validation set, bit by bit) assure that general models are favored.

Comment 10: I was totally lost in the philosophical arguments at the end of the introduction paper. Please delete, it appears to have been placed to impress the reader. I am reacting to it in quite an opposite manner

Reply 10: We agree that the discussion about the fundamental natural complexity (lines 98-112 in the manuscript) is not strongly connected to the rest of the manuscript. Nevertheless, we think it is important to mention that all the concepts discussed throughout the manuscript measure only facets of the complexity of the natural system under investigation. In a revised version of the manuscript, we will shorten this section.

Comment 11: The way Prob for entropy measure has been calculated is in itself a model that depends on the choice and number of bins. That has implications for how well Prob has been estimated from limited data in terms of how such frequency estimates converge to true Prob (ie it has its own complexity challenges) that the approach so very much relies on. Perhaps this can be discussed in bit more detail.

Reply 11: Agreed. We will add a short discussion about the influence of binning choices, and point to related literature.

Finally two major comments:

Comment 12: the authors should show predictive performance to demonstrate generalizability. Or validation, even if in narrative form, by comparing their conclusions with what other authors, not linked to information theory applied to water, have said.

Reply 12: Please see our replies to comment 8.

Comment 13: the authors again need to place their finding in the landscape of other complexity studies, especially in modeling MOPEX catchments, in hydrology. How do their conclusions regarding complexity compare with the narrative presented here? This will only add value to an already large literature set of hydrological model complexity, esp wrt to streamflow.

Reply 13: Agreed. The usages of the term 'complexity' are manifold – not only but also in hydrology - and to date no unique definition exists. For clarification, and to put our work into context, we will add to the manuscript a brief overview on its usage in hydrology, and refer more extensively to prior work on model complexity in hydrology.

Yours sincerely,

Uwe Ehret, on behalf of all co-authors

---

## Author Response (AR1)

**Cover letter to revision of hess-2020-128**

Dear Editor, dear Referees,

We have completed a comprehensive revision of the manuscript according to our suggestions made in the replies to the referee comments and the short comments. Below we indicate the changes made with respect to each comment; the related page and line numbers refer to the revised version of the manuscript. For brevity and clarity we do not repeat the referee and short comments and our replies in full length here. Please find them in HESSD. The changes in the manuscript are marked in red.

**RC 1 (Elena Toth)**

Comment 1: We have rearranged and mostly re-written the entire introduction (section 1) along the suggestions made in the replies to the referee. We have also mostly re-written the abstract for the same reason, and made some modifications to the summary and conclusion (section 4). We hope the updated version of the text is easier to read now, and better conveys the main messages of the paper.

Comment 2:

We have added a better explanation of the purpose and the key messages of the model comparison to section 1.3 in the introduction ('Scope and goals of this paper').

We have replaced the previous 10-node ANN by an LSTM with a single hidden layer of 5 neurons, which is more capable of reproducing a catchments storage-release behaviour. We hope the referee finds this a fairer treatment of the ANN class of models in our study. (see Table 1 on page 8). The LSTM shows better performance than the previous ANN, but the computational effort is still high.

We have changed our model application setup. Instead of training and evaluating the models on all available data, we now follow an validation set approach, training the models on the first half of the data, and evaluating them on the second half. The setup is explained in detail in the new introduction (section 1); all results, plots etc. have been updated accordingly.

Comment 3: We have re-written most of the abstract (please see also our reply to comment 1)

Comment 4: We have maintained the detailed discussion of the 'philosophical' background of the bit by bit method as we believe it is important to put the two contributions of the paper into a larger perspective. Please see also our reply to comment 1.

Comment 5: We moved Eq. 1 into Table 1 (page 8).

Comment 6: We fit the models in a calibration period, and present results for a validation period. Please see also our reply to comment 2.

Comment 7: We have added the entropy equation and a short explanation about how it is related to information and conditional entropy at the beginning of section 2.4.1 (page 9).

Comment 8: No changes made to the manuscript

**RC 2 (Anonymous)**

Comment 1: No changes made to the manuscript

Comment 2: Following our suggestions in the reply to the referee, we have adjusted the abstract (last paragraph) and the introduction (page 5 line 121 pp) ) to better explain what we mean by 'general'.

Comment 3: We have added to the introduction (page 4 line 80 pp) a sentence, briefly naming AIT as the basis for the claim of generalizability, and pointing to Weijs and Ruddell (2020) and references therein for more detail.

Comment 4: We have added a short description of the two possible ways to evaluate information in model predictions: As information loss against an upper benchmark (the observations), or information gain against a lower benchmark (the entropy of a uniform distribution) (page 9 lines 197-200, and page 10, lines 212-216). We also added a short explanation that we use information loss to remain consistent with Weijs and Ruddell (2020). (page 9 lines 214-216).

Comment 5: No changes made to the manuscript

Comment 6: Throughout the text, we have replaced 'universality' with 'generality', and we have adjusted the abstract (last paragraph) and the introduction (page 5 line 121 pp) ) to better explain what we mean by 'general'. Comment 7: No changes made to the manuscript

Comment 8: We have rearranged and mostly re-written the entire introduction (section 1) along the suggestions made in the replies to the referee. We have also mostly re-written the abstract for the same reason, and made some modifications to the summary and conclusion (section 4). We hope the updated version of the text is easier to read now, and better conveys the main messages of the paper.

Comment 9: We have changed our model application setup. Instead of training and evaluating the models on all available data, we now follow an validation set approach, training the models on the first half of the data, and evaluating them on the second half. The setup is explained in detail in the new introduction (section 1); all results, plots etc. have been updated accordingly.

Comment 10: We removed the entire paragraph about the relation between descriptive, computational, and natural complexity at the end of section 1.3.

Comment 11: We added a short discussion about the influence of binning choices to section 2.4.1 (page 10 lines 218-224).

Comment 12: Please see our reply to comment 9.

Comment 13: We have added a new section 1.4 at the end of chapter 1, where we give a short overview on uses of the term 'complexity' and related research in the hydrological sciences, and put our uses of the term into perspective.

**SC 1 (John Ding)**

Comment 1: In addition to the replies given in ReplySC1, we would like to add that, based on recommendations by John Ding, referee #1 and referee #2, we have made several changes with respect to the range of models used for demonstration purposes: All models are now calibrated on the first half of the available data; results in terms of performance and computational complexity are from applying

the models on the previously unseen second half of the data. For calibration, we used the well-known Nash-Sutcliffe efficiency (NSE). We hope that these changes, which follow standard hydrological practice, will help the target audience – hydrologists – to better connect to the approach and key messages of the paper. Nevertheless, the purpose of applying a range of models in this paper is not to compare their performance; the key messages are that

- computational complexity as measured by 'strace' is sensitive to all computational aspects of a model: the amount of required forcing data, the size of the model itself, its time stepping, spatial resolution, numerical scheme, etc. ,
- 'strace' can be applied to any computer-based model, i.e. it is general within its range of application (evaluation of computer-based models).

We added a sentence explaining this in the last paragraph of the abstract, and the introduction (page 5 line 121 pp).

Comment 2: No changes made to the manuscript

Comment 3: No changes made to the manuscript

**SC 3 (John Ding)**

Comment 1: No changes made to the manuscript

**SC 4 (John Ding)**

Comment 1: No changes made to the manuscript

**SC 2 (Robert Reinecke)**

Comment 1: We have added to the summary & conclusions the phrase " in the general setting of incremental learning" (page 16 line 360).

Comment 2: We have added a sentence plus a reference to Hutton et al. (2016) to the summary and conclusion (page 16 lines 364-367).

Comment 3: We changed the related sentence in the abstract (page 1 lines 15 pp).

Comment 4: We changed the related sentence in the abstract (page 1 lines 15 pp).

Yours sincerely,

Uwe Ehret, on behalf of all co-authors

---

## Author Response (AR2)

**Cover letter to second revision of hess-2020-128**

Dear Editor, dear Referees,

Please find our replies to the second round of reviews and the related Editor comment below. We provide a point-by-point reply and indicate changes made to the manuscript. The related page and line numbers refer to the second revised version of the manuscript. The Editor and Referee comments in this document are marked in blue, in the manuscript all changes are marked in red.

**Editor note**

Comment 1: Dear Authors, as you can see, both Referees recommend major revisions. Referee #1 Elena Toth still casts doubts on the way the comparison among the models is carried out. Indeed, although you argue that the bit-by-bit approach you propose allows comparing models which make use of different input data, I still see Elena's point: if two modelling approaches CAN make use of the same data, comparing them with different inputs by means of an index that accounts (also) the obtained performance can be misleading.

Reply 1: As suggested by Elena Toth, we added a simple ANN with the same input as the autoregressive model (Q(t-1), Q(t-2), Q(t-3)) to the set of models used in the paper, and now provide a more in-depth discussion about how the bit-by-bit method can be used to guide model optimization and model selection in sections 3.1 and 3.2. Please also see our related reply to Comment 1 by Elena Toth.

Comment 2: The other Referee #2 raises more fundamental issues about the way you present your approach. A less narrow review of the wide literature about model parsimony would help the reader to better place your contribution in the context of existing research on the topic.

Reply 2: Agreed. As model parsimony and model complexity are strongly linked (parsimonious models are models of minimally adequate complexity), we have decided to add an overview on related literature to section 1.4 (uses of 'complexity' in the hydrological sciences). Also, we created in section 1.2 (Guidelines for developing parsimonious models) a new paragraph about 'Model selection by applying complexity penalization measures' and added this to Figure 1 (b), as i) these approaches are often used and ii) can be considered a particular class of methods.

Comment 3: And also, to better support your claim about the approach being capable of guiding in model optimization, a clearer training-validation approach to your performance comparison would be probably useful, as suggested by Referee #2.

Reply 3: We agree that the manuscript will benefit from better explaining how the bit-by-bit method can be used to guide model optimization and model selection. We think this can be best done by discussing several typical use cases of model optimization and model selection. We now do so in section 3.2. Please see a detailed description of the use-cases in our reply to Comment 1 by Elena Toth.

Comment 3: Given all this, I definitely agree that major revisions are still required, and I invite you to take seriously into consideration all the raised issues, providing a convincing rebuttal in case you still think that some of them have not to be addressed. I look forward to receiving a new version of your manuscript. Best regards, Roberto Greco

Reply 3: We hope that with our replies and changes made to the manuscript, we could address the Editor's and the Referees' concerns in a satisfactory manner.

**RC 1 (Elena Toth)**

Comment 1: Dear Authors, I do appreciate that you are now presenting the results of a calibration/validation split-sample test and you improved the manuscript in many points. I am also glad that you have better explained the contribution of the paper and highlighted that "It is important to note that the purpose of the model comparison here is not primarily to identify the best among the different modelling approaches", since actually I keep thinking that for a fair comparison of the performance of the models you cannot use different input. When you write (in the discussion) "It is therefore neither required nor useful to establish models that all rely on the same input data, because the bit by bit approach might be used for exactly the purpose of comparing two models relying on different input data", I think that this holds true for the complexity measure but not for the performance one. In fact, the correct input strongly influence the 'side information' that is needed to reduce the information loss, since the input information is crucial to the information content embedded in the model prediction. As written in Campolo (WRR, 1999), the last observed discharges are generally included as inputs to ANN runoff-prediction models, since they provide information on the state of saturation of the basin, which is a function of the history of the meteorological input in the period preceding the streamflow generation: the capability of the system to respond to rainfall perturbation is represented by the ongoing streamflow measured in real-time at the closing section. And in fact you use just such input for the best performing model (#7). Such model is by far the best performing one since it is the only one that uses such a precious information (and actually also a fair comparison of the performances of the conceptual models would require, see Toth and Brath 2007, a real-time updating procedure making use of such data). For this reason, the new ANN you applied, still fed by rainfall only, still provides a relatively poor performance in comparison to Model 7 (and the complexity seems to be even worse than in the first version). If, instead of using your complex LSTM fed by the rainfall values, you had used a simple multi-layer ANN (again see what I did in Toth and Brath, WRR, 2007) with the same input (past Q values) that you use with the autoregressive model, you would have got performances similar (or probably better) than Model 7, even if the complexity would be, I guess, still much higher than that of Model 7. Such comparison would be not only fairer, but also more significant than what you are presenting now. Such a simple ANN model implementation would take very little energy for you to set up, and, respectfully, I would insist for you to do so, since it would provide sounder results, supporting the scientific relevance of your interesting approach. Elena Toth

Reply 1: We are glad that Elena Toth acknowledges most changes we made to the manuscript and accept her request for providing a fairer model comparison by using the same input for the models that are compared. In fact, we had the same intention when we decided to provide model 8 (the LSTM) with the same input data (precipitation) as the bucket models, to allow for a fair comparison among these. Elena Toth correctly states that this neglects all the information available in past observations of discharge, which are used by model 7 (the autoregressive model), and therefore provides an advantage for the AR-model when comparing it to the LSTM. To resolve this issue, and to also better demonstrate how the bit-by-bit method can be used to guide model optimization and model selection, we

- added a simple ANN (single hidden layer with five neurons) with the same input as the autoregressive model ($Q(t-1)$, $Q(t-2)$, $Q(t-3)$) to the set of models used in the paper, and kept the LSTM
- we have completely re-written section 3.2 and now discuss several use cases of model optimization and model selection in section 3.2.

We also acknowledge Elena Toth's remark about "the input information being crucial to the information content embedded in the model prediction" by adding a related sentence in the summary and conclusions (P18 L425)

**RC 2 (Anonymous)**

Comment 1: I have gone through the revised version of the manuscript. Unfortunately the authors either have evaded almost all my comments (except putting the paper in the landscape of select complexity studies) or have decided to respond to comments of their own. They have accentuated their reliance on Weijs and Ruddell without providing either theoretical or empirical support for their claims. Following are some of my comments (only on the added text in red by the authors)

Reply 1: We would like to emphasize that we addressed in a point-by-point manner all comments made by the referee, and we justified each case where we did not follow the referee suggestions.

The referee mentions we responded to comments of our own. As the referee does not provide detail information about where this is the case, we assume that these are cases where we responded to comments made by the other referee.

The referee mentions the strong linkages of the manuscript to Weijs and Ruddell (2020). We are not sure what to do with this comment. Yes, there is a strong connection of this manuscript to Weijs and Ruddell (2020), and throughout the manuscript we explain how they are related to each other (especially in section 1.2), but we do not understand why this should be a problem.

The referee mentions that we do not provide theoretical or empirical support for our claims. As there is no detail information about which claims the referee refers to, we assume it is linked to referee comment 3 and comment 6 of the first round of reviews:

- Comment 3: ("How max parsimony + max performance -> max generalizability? Theoretical rigor behind the claim is missing").
- Comment 6: (" … Author's claim to universality should first provide a rigorous theoretical treatment that has not even been provided in the WRR paper that the authors allude to").

With respect to these comments, we responded that for better readability of the hydrological readership of HESS, we refer to Weijs and Ruddell (2020) and related literature about AIT rather than including it into our manuscript. Following the referee's repeated request, we have now included a brief discussion on the theoretical foundations of our claims in section 1.2 (a detailed list of additions is provided at the end of this reply)

Re comment 3: Adding some form of performance measure and some form of complexity penalization is a widely accepted way to do model selection (which often has the underlying aim of inference or prediction, both of which are related to generality). See for example AIC and BIC information criteria, statistical learning theory, algorithmic information theory, and the minimum description length principle. We have now made this clearer in the paper by adding some discussion on AIC and BIC and how they relate to the concepts we discuss.

Re comment 6: Since proofs related to AIT rely on a wide range of prior knowledge about theoretical computer science, we see little point in repeating those in a paper addressed at an audience of hydrologists. To the long explanation in the previously reply to the reviewer, which we did not include in the paper for the reasons above, we could add the following: Since the space of all self-delimiting programs for a universal Turing machine form a prefix-free code (no valid program is the first part of

another valid program), a program that is one bit longer than another program will be half as likely to be encountered in random noise on the input tape. Therefore, the algorithmic probability of a certain output sequence is proportional to the probability of encountering that output string from a Turing Machine fed with random noise.

We honestly believe adding this to the paper would serve little purpose for the reader. We therefore kept our descriptions at an intuitive level, without too much computer science terminology, but elaborated a bit more on those in the present paper.

Related changes made to the manuscript:

- P4 L68-72: We created in section 1.2 (Guidelines for developing parsimonious models) a new paragraph about 'Model selection by applying complexity penalization measures' and added this to Figure 1 (b). In this paragraph, we discuss AIC and BIC in more detail
- P4 L74-78, P4 L93-96: Added some more information about AIT
- P4 L87-88: Added some more information about the approach of Weijs and Ruddell (2020)

Comment 2: Please also discuss the work of hydrological complexity by Sivapalan, Wagener and colleagues. Where/how does your paper compare and why if not?

Reply 2: We have added related literature and a discussion to section 1.4 (P7 L173-175). The main difference between the work on complexity of Sivapalan, Wagener and colleagues and what we propose in our manuscript is in the definition of 'complexity'. We also added more literature about ways to define and measure complexity (P7 L165-168).

Comment 3: "Weijs and Rudell (2020), we express model performance in terms of information losses" is limited by how p(.) is estimated (it being prone to misspecification) in Equation 2. This in itself is a model, subject to complexity associated challenges. I would therefore suggest caution in suggesting Weijs and Rudell (2020) at the same level of rigour as some other fanstastic work done on Occam's razor in Mathematical Physics and Applied Probability. For example, no underlying arguments for uniform convergence, asymptotic consistency etc are provided other than authors guiding us to literature on information theory.

Reply 3: The point about estimation of p being a model was raised by the referee in comment 11 of the first round of reviews. We agreed and added a discussion in section 2.4.1. In terms of convergence, several estimators for discrete distributions based on limited samples have been proposed that both converge asymptotically towards the true distribution and provide uncertainty bounds as a function of sample size and binning choice. In Darscheid et al. (2018), both a Bayesian approach and a Maximum-Likelihood approach are presented. We added this reference and a short explanation to the manuscript (P11 L246-249).

As for Weijs and Ruddell (2020), in that case information loss is expressed as the minimum description length needed to reproduce the observations. As such, if probability is used, the model to describe p (e.g. the storage of the histogram) is included in the description length and a misspecification or overfitting of the probability model is penalized automatically. One key philosophical advantage of AIT is that the information measures do not refer to an underlying probability distribution, but works directly on the observed data. This gain in generality and rigor is paid for by less practical applicability. In this paper, we do not consider descriptive complexity of the model nor the probability model, and therefore use classic entropy measure in combination with binning approaches recommended in the literature. We agree this is an area worthy of further future research. Since this is again a quite complex and subtle

discussion to explain well in the paper, and not directly within the scope of the central points of the present paper, we decided not to elaborate on this in the paper.

Comment 4: What are the implications of measuring information gains compared to a lower benchmark – the entropy of a uniform distribution, in terms of quantifying model performance. Is it an approximation? Or just some measure unrelated to approximation error (which I think is the case)?

Reply 4: Upper and lower benchmarks are helpful to put a particular model performance into a global perspective. For example, Nash-Sutcliffe efficiency (NSE) is limited to values $[-\infty \ 1]$. Models showing NSE-values smaller than zero (the NSE of the mean as a lower benchmark model) are typically rejected, and hydrological models showing NSE-values > 0.8 are typically considered well-performing. So clearly there is some merit in knowing general upper and lower bounds of NSE to put the performance of a particular model into a wider perspective. The same is the case for information gain, where the entropy of a uniform distribution provides a global lower bound. The difference between the two lower bounds is that for NSE it is a function of the data under investigation (as we use them when calculating the mean), and for entropy it is a function of the resolution (number of bins the value range is divided in) for which we investigate the agreement of observation and simulation. We can interpret the entropy as the missing information about which bin (at the chosen precision) the value falls in, when we have a priori knowledge of the range of the data.

Comment 5: "Choice of $n$ is typically guided by the objective to balance resolution and sufficiently populated bins" – exactly and this somehow sounds like Occam's razor again. See my comment above.

Reply 5: Yes, choice of the number of bins can be seen as an optimization problem (see e.g. the binning approach suggested by Knuth (2013). Please also see our replies to comment 3.

Comment 6: Not clear what validation set approach actually doing. I would like to see how model chosen based on the principle presented here performs on the validation set (if it somehow identifies a generally good performing models) in comparison with other 'demoted' models. Otherwise its just a trivial exercise, sure you can have very simple but really bad models. We do not need to understand it bit by bit.

Reply 6: We do not understand what is unclear about the validation set approach as we explain it in section 2.1, and which is standard practice in hydrological modelling. We hope we could clarify this issue by the use cases of model optimization and model selection that we now discuss in section 3.2.

Comment 7: "We continued by describing several paradigms to guide model development:.. Weijs and Ruddell (2020) express both model performance and descriptive complexity in bit, and by adding the two obtain a single measure for what they call 'strong parsimony';.." that ok, but the authors are not the only ones attempting to describe parsimony. Don't even think Weijs and Ruddell (2020) is a paradigm. Please delete or list many others who have dealt with concepts of parsimony.

Reply 7: Following the suggestion by the Referee and the Editor, in section 1.2 (Guidelines for developing parsimonious models), we now give a short overview on approaches to develop parsimonious models in addition to those that are already referred to and used in the paper (Occam's razor, complexity penalization, strong parsimony as suggested by Weijs and Ruddell (2020), validation set, and bit by bit).

Comment 8: "Occam's razor puts an emphasis on descriptive complexity, considers performance as a side condition, but it ignores computational complexity;.." this is incorrect interpretation (on performance being side condition). Please delete in order not to misguide readers.

Reply 8: As described in Weijs and Ruddell (2020), and repeated in our manuscript (P X L X), "Occam's Razor, a bedrock principle of science, argues that the least descriptively complex model is preferable, at a given level of predictive performance that is adequate to the question or application at hand." Predictive performance is used as a threshold to select a set of models, which are then compared in terms of descriptive complexity, and the model with least descriptive complexity is then selected as the best. Predictive performance here serves as a threshold-like filter, which we wanted to express by 'side condition'. As this was apparently misleading, we rephrased the sentence the referee refers to:

"When applying Occam's Razor, the parsimonious among the well-performing models are identified, but comparisons of models of different complexity for model selection are not possible by this principle alone." (P4 L65-67).

"Occam's razor puts an emphasis on descriptive complexity, and is often combined with performance considerations, but it ignores computational complexity;" (P17 L395).

Comment 9: "i) measuring computational complexity by 'Strace' is general in the sense that it can be applied to any model that can be run on a digital computer; ii) 'Strace' is sensitive to all aspects of a model, such as the size of the model itself, the input data it reads, its numerical scheme and time-stepping; iii) the 'bit by bit' approach is general in the sense that it measures two key aspects of a model in the single unit of bit, such that they can be used together to guide model analysis and optimization in a pareto trade-off manner in the general setting of incremental learning. It can be useful.." In general I agree with i). I find their iii) statement misguided – I don't think strace can be used for incremental learning, since computational complexity doesnot make us unlearn something. "Descriptive complexity'" may do so (not based on bit by bit idea however) if interpreted appropriately in how it affects learning. Therefore computational complexity is a redundant concept in context of learning.

Reply 9: We agree that if the computational effort of a model is not an issue, computational complexity is indeed not an important criterion for model optimization and model selection. However, there are many applications in the Earth Sciences where computational effort is critical (e.g. global circulation models or global land surface models), and for which a lot of time and effort is devoted to increase their computational efficiency. In such a setting, computational complexity can be an important criterion for model evaluation. E.g. consider two land surface models, which are identical except for their spatial resolution. If the two models are equal in terms of performance, but the coarser-resolution model is computationally  more efficient, we will prefer this model, and we have at the same time learned something about the adequate resolution to represent the natural system in the model. In that sense computational complexity also plays a role in learning. We hope this is also made clearer in the manuscript by adding the use cases in section 3.2.

Comment 10: Over all, I think another revision is at least needed where Weijs and Rudell (2020) is not glamorized (and reference to it kept to a minimum) since all arguments based on it are rather weak.

Reply 10: There is a strong connection of this manuscript to Weijs and Ruddell (2020), and throughout the manuscript we explain how they are related to each other (especially in section 1.2), but we do not understand why this should be a problem (please see also our reply to Comment 1).

Comment 11: The authors should clearly show how computational complexity (or even Weijs and Ruddell description complexity) is affecting learning based on performance (only) of all the models on the validation set. My contention here is that computational complexity in this regard (of generalized learning) is rather of little value and that the paper is merely another demonstration of an interesting concept that does not add much to learning from data.

Reply 11: We hope we addressed this comment appropriately by our replies to Comment 6 and 9, and by adding to the manuscript the discussion of use cases in section 3.2.

Comment 12: I would advise humble response in the next iteration, where limitations and purpose of their manuscript is clearly documented.

Reply 12: With all respect, we would like to mention that while we welcome the scientific debate associated with this review process, and appreciate the time and effort spent by the referee, to our understanding the scientific debate is not to supposed to be humble but rather respectful and professional. Therefore, we aim to convey neither arrogance not humbleness but solely the respectful, professional civility of a scientific debate. That said, we did add an elaborate section on the use cases of the method to clarify its purpose. We think limitations are fairly discussed, and we hope the manuscript will contribute to the scientific discussion.

Yours sincerely,

Uwe Ehret, Elnaz Azmi, Steven Weijs, Benjamin Ruddell, Rui Perdigao

**References**

Darscheid, P., Guthke, A., and Ehret, U.: A Maximum-Entropy Method to Estimate Discrete Distributions from Samples Ensuring Nonzero Probabilities, Entropy, 20, Article:-601, 10.3390/e20080601, 2018.

Gupta, H. V., Wagener, T., and Liu, Y.: Reconciling theory with observations: elements of a diagnostic approach to model evaluation, Hydrol. Process., 22, 3802-3813, 10.1002/hyp.6989, 2008.

Knuth, K.: Optimal Data-Based Binning for Histograms, arXiv:physics/0605197v2 [physics.data-an] 2013.

Weijs, S. V., and Ruddell, B. L.: Debates: Does Information Theory Provide a New Paradigm for Earth Science? Sharper Predictions Using Occam's Digital Razor, Water Resources Research, 56, e2019WR026471, 10.1029/2019wr026471, 2020.